# Dark Matter EFT, the Third — Neutrino WIMPs

Ingolf Bischer[1][⋆], Tilman Plehn[2][†] and Werner Rodejohann[1][‡]

**1** Max-Planck-Institut für Kernphysik, Saupfercheckweg, Heidelberg, Germany
**2** Institut für Theoretische Physik, Philosophenweg, Universität Heidelberg, Germany

⋆ bischer@mpi-hd.mpg.de, † plehn@uni-heidelberg.de, ‡ werner.rodejohann@mpi-hd.mpg.de

## Abstract

Sterile neutrinos coupling only to third-generation fermions are attractive dark matter candidates. In an EFT approach we demonstrate that they can generate the observed relic density without violating constraints from direct and indirect detection. Such an EFT represents a gauged third-generation $B-L$ model, as well as third-generation scalar or vector leptoquarks. LHC measurements naturally distinguish the different underlying models.



# 1 Introduction

What particles form dark matter (DM), how do they couple to the Standard Model (SM), and how did they get produced? These questions require a fundamental interpretation framework for a stable new particle from a new physics sector. After defining its hypothetical quantum numbers we can add the dark matter particle to the Lagrangian of the SM and obtain a perfectly consistent theory. If this Lagrangian is perturbatively renormalizable, the couplings to the SM are given by the quantum numbers of the dark matter agent. If the Lagrangian includes higher-dimensional operators, the interaction through a decoupled new mediator defines a dark matter effective theory. The only constraint we have on these interactions is the measured relic density, combined with an assumed production mechanism in the thermal history of the Universe.

Of many candidates and scenarios, weakly interacting massive particles (WIMPs) [1–5] with their defining thermal freeze-out production still stand out [6,7]. The original papers all use sterile neutrinos as actual WIMPs, and we will return here to this notion. Direct detection experiments are slowly covering the relevant WIMP parameter space [8, 9]. To obtain the observed relic density from freeze-out production, the dark matter annihilation rate to SM-particles has to be large. Hence, SM-mediators like the $Z$-boson or even the Higgs boson are ruled out over almost the entire parameter space. New mediators can avoid these constraints, and if the annihilation process is not enhanced by an $s$-channel funnel it can be described by a dark matter EFT (DMEFT) [10–15]. The effective Lagrangian can be informally written as $\mathcal{L} = C/\Lambda^2 f^2 \text{DM}^2$, where $f$ is a SM fermion and DM the dark matter particle. We can then translate the observed relic density $\Omega_{\text{DM}} h^2 \approx 0.12$ into a typical dark matter annihilation cross section [16]

$$\langle \sigma_{\text{ann}} v_{\text{rel}} \rangle \overset{!}{\approx} \frac{1}{600 \text{ TeV}^2} \sim \frac{C^2 m_{\text{DM}}^2}{4\pi \Lambda^4}. \tag{1}$$

The relation between the dark matter mass and the new physics scale becomes

$$\frac{\Lambda^2}{C \, m_{\text{DM}}} = 6.9 \text{ TeV} \qquad \overset{m_{\text{DM}}=10 \text{ GeV}}{\Longrightarrow} \qquad \Lambda = 260 \text{ GeV} \sqrt{C}$$

$$\overset{m_{\text{DM}}=\Lambda/R}{\Longrightarrow} \qquad \Lambda = 6.9 \text{ TeV} \frac{C}{R}. \tag{2}$$

This indicates that large mass ratios $R = \Lambda/m_{\text{DM}}$ require light new physics to produce the correct relic densities, while large Wilson coefficients $C$ can alleviate this pressure. Too small values of $R$ will cast doubt on the EFT assumption, but we can assume a typical range

$$R = \frac{\Lambda}{m_{\text{DM}}} = 3 \dots 10 \qquad \text{and} \qquad C \gtrsim 1. \tag{3}$$

This picture of the DMEFT is consistent and describes not only the relic density, but also direct and indirect detection, typically leading to a lower bound on the DM mass, which can be more or less restrictive depending on the nature of the DM particle and the operators considered [17, 18]. However, the fact that the typical EFT-scales around 1 TeV which predict the correct relic abundance of DM for the traditional WIMP mass region around 100 GeV are directly probed by the LHC leads to a breakdown of the EFT and the need to consider explicit mediators unless one is willing to consider a strongly interacting regime [19–21]. This is one strong motivation for the application of simplified models instead of EFTs to new physics searches at the LHC [22]. Another possible way to save elements from the EFT is to include a mediator into the excitable degrees of freedom of the DMEFT [23, 24].

Our approach is that we keep the original DMEFT at the cost of having to resort to the following treatment at collider scales. Matching to the full theory we trade the new physics scale $\Lambda$ and the Wilson coefficient $C$ for mediator mass and coupling, $\Lambda \approx m_{\text{med}}$ and $C^2 \approx g^4$. This matching allows us to include collider measurements in a global analysis, even if these measurements require an on-shell production of the mediator. At the same time, matching to a full theory brings in constraints from the construction or the consistency of the model. For instance one needs to address flavor observables or assure perturbatively small values of $g$ whenever the model should be valid to high scales. If there exists no UV-complete theory completing a consistent EFT approach, such an EFT is utterly pointless. From this perspective, a fully consistent EFT approach is less trivial than one may think [19,20,25–27]. However, even if a straightforward construction of a UV-completion fails, it is difficult to rule out its existence. For instance, strongly interacting theories can lead to non-trivial EFT mappings [21].

In this paper, we build on the recent appearance of 4-fermion effective interactions in the neutrino sector [28–32], beyond oscillation analyses [33–36] and consider a DMEFT approach to a sterile neutrino. We let DM couple only to the *third* fermion generation, which naturally leads to suppressed DM-nucleon couplings and production cross sections at the LHC. First, considering the relevant higher-dimensional operators, we demonstrate that the relic density via freeze-out can be generated, while at the same time direct and indirect detection limits can be obeyed without losing the validity of the EFT approach. Next, we investigate possible UV-completions of this framework:

- the anomaly-free gauged Abelian $(B-L)_3$ symmetry. The dark matter sterile neutrino couples to a $Z'$ which in turn couples to third generation fermions;
- a scalar third generation leptoquark, which couples to the dark matter sterile neutrino and right-handed top quarks;
- a vector third generation leptoquark, which couples to the dark matter sterile neutrino and right-handed bottom quarks.

These models face different constraints in particular from LHC searches, while the direct and indirect detection constraints are accurately obtained within the corresponding EFT limit.

The paper is built as follows: In Section 2 we discuss the EFT of fermion singlets, which includes a variety of higher-dimensional operators. Focusing on couplings to third generation particles, we identify the operators relevant for DM freeze-out, as well as direct and indirect detection. We shown that our EFT scenarios allow for the successful DM generation in agreement with all constraints. In Section 3 we discuss some issues with consistent EFT scenarios. We then match our EFTs with explicit models, specifically the gauged Abelian $(B-L)_3$ as well as scalar and vector leptoquarks. The models are confronted with additional constraints from colliders and flavor, before we conclude in Section 4.

## 2 $\nu$DMEFT framework

For our $\nu$DMEFT we assume a sterile neutrino $N$ of mass $m_N \gtrsim 10$ GeV as the DM agent. Its mixing with active neutrinos must be very small or forbidden by symmetry, for instance if $N$ is the only particle odd under a $\mathbb{Z}_2$ symmetry. This forbids in particular a Yukawa coupling with the SM Higgs, which would make it unstable [37,38]. For our present purposes then, our $\nu$DMEFT is equivalent to the regular DMEFT. We simply identify the fermionic singlet DM as a stable (or at least very long-lived) sterile neutrino to stress the connection to the currently actively discussed topic of Standard Model Effective Field Theory (SMEFT) extended with sterile neutrinos ($\nu$SMEFT or SMNEFT) [35,39–45]. The standard seesaw mechanism as one possible origin of Majorana neutrinos $N$ implies that other sterile neutrinos would have to be heavier, so they decouple and decay before $N$ freezes out. For instance, two heavy

sterile Majorana neutrinos are sufficient for the seesaw mechanism to generate light neutrino masses (one of which would be massless), and leave a super GeV-scale sterile neutrino as a DM candidate. However, this realization is only one possibility, there are countless ways to generate active neutrino masses. We will also consider the Dirac option for $N$ in the course of the paper, and will comment on the difference to the Majorana case when appropriate.

## 2.1 Operator basis

We describe a sterile Majorana neutrino as a 4-component spinor

$$N = \begin{pmatrix} n_R^c \\ n_R \end{pmatrix} = N_R^c + N_R \,, \tag{4}$$

where $N_R$ and $N_R^c$ are 4-component chirality eigenstates. Assuming only SM gauge symmetries, the renormalizable operators are given by the SM-Lagrangian plus a sterile neutrino contribution,

$$\mathcal{L}_4 = \mathcal{L}_{\text{SM}} + i\overline{N}_R \gamma^\mu \partial_\mu N_R + \left( \frac{1}{2} m_N \overline{N_R^c} N_R + \text{h.c.} \right) + \left( \bar{l} Y_{lN} N_R \tilde{H} + \text{h.c.} \right) . \tag{5}$$

We extend this Lagrangian by D5 and D6 terms,

$$\mathcal{L}_{\text{eff}} = \mathcal{L}_4 + \mathcal{L}_5 + \mathcal{L}_6 = \mathcal{L}_4 + \frac{1}{\Lambda} \sum_i C_i \mathcal{O}_i^{(d=5)} + \frac{1}{\Lambda^2} \sum_i C_i \mathcal{O}_i^{(d=6)} . \tag{6}$$

Table 1: 4-fermion D6 operators in the SMEFT (upper) and after adding right-handed neutrino singlets (lower) to generate the $\nu$SMEFT. Note that $l_\alpha$ and $q_\alpha$ denote the weak lepton and quark doublets of flavor $\alpha$, individual fermions $u_\alpha$, $d_\alpha$, and $e_\alpha$ are the weak singlets. Regarding the operator $\mathcal{O}_{ff'}$, the index $ff'$ runs over $ee, uu, dd, eu, ed$, and $ud$ of all 3 generations. Flavor indices of the operator symbols are suppressed in the table. This list does not include baryon or lepton number violating operators.

| $(\overline{L}L)(\overline{L}L)$ and $(\overline{R}R)(\overline{R}R)$ | | $(\overline{L}L)(\overline{R}R)$ | | $(\overline{L}R)(\overline{R}L)$ and $(\overline{L}R)(\overline{L}R)$ | |
|---|---|---|---|---|---|
| $\mathcal{O}_{ll}$ | $(\bar{l}_\alpha \gamma_\mu l_\beta)(\bar{l}_\gamma \gamma^\mu l_\delta)$ | $\mathcal{O}_{le}$ | $(\bar{l}_\alpha \gamma_\mu l_\beta)(\bar{e}_\gamma \gamma^\mu e_\delta)$ | $\mathcal{O}_{elqd}$ | $(\bar{e}_\alpha l_\beta^j)(\bar{q}_\gamma^j d_\delta)$ |
| $\mathcal{O}_{lq}^{(1)}$ | $(\bar{l}_\alpha \gamma_\mu l_\beta)(\bar{q}_\gamma \gamma^\mu q_\delta)$ | $\mathcal{O}_{lu}$ | $(\bar{l}_\alpha \gamma_\mu l_\beta)(\bar{u}_\gamma \gamma^\mu u_\delta)$ | $\mathcal{O}_{eluq}$ | $(\bar{e}_\alpha l_\beta^j)\epsilon_{jk}(\bar{u}_\gamma q_\delta^k)$ |
| $\mathcal{O}_{lq}^{(3)}$ | $(\bar{l}_\alpha \gamma_\mu \tau^I l_\beta)(\bar{q}_\gamma \gamma^\mu \tau^I q_\delta)$ | $\mathcal{O}_{ld}$ | $(\bar{l}_\alpha \gamma_\mu l_\beta)(\bar{d}_\gamma \gamma^\mu d_\delta)$ | $\mathcal{O}'_{eluq}$ | $(\bar{e}_\alpha \sigma_{\mu\nu} l_\beta^j)\epsilon_{jk}(\bar{u}_\gamma \sigma^{\mu\nu} q_\delta^k)$ |
| $\mathcal{O}_{qq}^{(1)}$ | $(\bar{q}_\alpha \gamma_\mu q_\beta)(\bar{q}_\gamma \gamma^\mu q_\delta)$ | $\mathcal{O}_{qe}$ | $(\bar{q}_\alpha \gamma_\mu q_\beta)(\bar{e}_\gamma \gamma^\mu e_\delta)$ | $\mathcal{O}_{quqd}^{(1)}$ | $(\bar{q}_\alpha^j u_\beta)\epsilon_{jk}(\bar{q}_\gamma^k d_\delta)$ |
| $\mathcal{O}_{qq}^{(3)}$ | $(\bar{q}_\alpha \gamma_\mu \tau^I q_\beta)(\bar{q}_\gamma \gamma^\mu \tau^I q_\delta)$ | $\mathcal{O}_{qu}$ | $(\bar{q}_\alpha \gamma_\mu q_\beta)(\bar{u}_\gamma \gamma^\mu u_\delta)$ | $\mathcal{O}_{quqd}^{(8)}$ | $(\bar{q}_\alpha^j T^A u_\beta)\epsilon_{jk}(\bar{q}_\gamma^k T^A d_\delta)$ |
| $\mathcal{O}_{ff'}$ | $(\bar{f}_\alpha \gamma_\mu f_\beta)(\bar{f}'_\gamma \gamma^\mu f'_\delta)$ | $\mathcal{O}_{qd}$ | $(\bar{q}_\alpha \gamma_\mu q_\beta)(\bar{d}_\gamma \gamma^\mu d_\delta)$ | | |
| $\mathcal{O}_{ud}^{(8)}$ | $(\bar{u}_\alpha \gamma_\mu T^A u_\beta)\times$ | $\mathcal{O}_{qu}^{(8)}$ | $(\bar{q}_\alpha \gamma_\mu T^A q_\beta)(\bar{u}_\gamma \gamma^\mu T^A u_\delta)$ | | |
| | $(\bar{d}_\gamma \gamma^\mu T^A d_\delta)$ | $\mathcal{O}_{qd}^{(8)}$ | $(\bar{q}_\alpha \gamma_\mu T^A q_\beta)(\bar{d}_\gamma \gamma^\mu T^A d_\delta)$ | | |
| $\mathcal{O}_{Ne}$ | $(\overline{N}_\alpha \gamma_\mu N_\beta)(\bar{e}_\gamma \gamma^\mu e_\delta)$ | $\mathcal{O}_{Nl}$ | $(\overline{N}_\alpha \gamma_\mu N_\beta)(\bar{l}_\gamma \gamma^\mu l_\delta)$ | $\mathcal{O}_{Nlel}$ | $(\overline{N}_\alpha l_\beta^j)\epsilon_{jk}(\bar{e}_\gamma l_\delta^k)$ |
| $\mathcal{O}_{Nu}$ | $(\overline{N}_\alpha \gamma_\mu N_\beta)(\bar{u}_\gamma \gamma^\mu u_\delta)$ | $\mathcal{O}_{Nq}$ | $(\overline{N}_\alpha \gamma_\mu N_\beta)(\bar{q}_\gamma \gamma^\mu q_\delta)$ | $\mathcal{O}_{lNqd}$ | $(\bar{l}_\alpha^j N_\beta)\epsilon_{jk}(\bar{q}_\gamma^k d_\delta)$ |
| $\mathcal{O}_{Nd}$ | $(\overline{N}_\alpha \gamma_\mu N_\beta)(\bar{d}_\gamma \gamma^\mu d_\delta)$ | | | $\mathcal{O}'_{lNqd}$ | $(\bar{l}_\alpha^j \sigma_{\mu\nu} N_\beta)\epsilon_{jk}(\bar{q}_\gamma^k \sigma^{\mu\nu} d_\delta)$ |
| $\mathcal{O}_{NN}$ | $(\overline{N}_\alpha \gamma_\mu N_\beta)(\overline{N}_\gamma \gamma^\mu N_\delta)$ | | | $\mathcal{O}_{lNuq}$ | $(\bar{l}_\alpha^j N_\beta)(\bar{u}_\gamma q_\delta^j)$ |
| $\mathcal{O}_{eNud}$ | $(\bar{e}_\alpha \gamma_\mu N_\beta)(\bar{u}_\gamma \gamma^\mu d_\delta)$ | | | | |

At dimension five the SMEFT [46] only features the Weinberg operator, while the neutrino extension allows for two additional terms [47],

$$\mathcal{L}_5 = \frac{C_{\nu\nu}}{\Lambda}\,(\overline{l^c}i\sigma_2 H)M_\nu(\widetilde{H}^\dagger l) + \frac{C_{NH}^{(5)}}{\Lambda}\,\overline{N_R^c}N_R\,H^\dagger H + \frac{C_{NB}^{(5)}}{\Lambda}\,\overline{N_R^c}\sigma_{\mu\nu}N_R\,B^{\mu\nu} + \text{h.c.} \qquad (7)$$

Here $B_{\mu\nu}$ is the hypercharge field strength. For only one Majorana neutrino, the magnetic-dipole-like $\mathcal{O}_{NB}^{(5)}$ is forbidden by the antisymmetry of the tensor-fermion bilinear. All three D5 terms violate lepton number by two units. Below the weak scale, the operator $\mathcal{O}_{NH}^{(5)}$ generates a mass correction for the sterile neutrino, but also Higgs (portal) couplings of the vertex forms $NNh$ and $NNhh$. Whenever a pair of sterile neutrinos may annihilate into a SM fermion pair, this operator is also produced at one-loop level. This is the case for all models studied in this work. Therefore, we will occasionally return only to $\mathcal{O}_{NH}^{(5)}$ out of the D5 operators in the following.

An operator basis at dimension six, including sterile Majorana neutrinos, is discussed in Refs. [48] and [49], see also Ref. [35]. We reproduce the Warsaw basis in Table 1 for the 4-fermion operators and Table 2 for mixed fermion-boson operators. In the upper sections of the tables we list the SMEFT operators, while in the lower sections we add the operators involving sterile neutrinos ($\nu$SMEFT). We leave out operators not connected to fermions, as well as lepton or baryon number violating operators unless originating from $N$, and the operator $\mathcal{O}_{N^4} = \overline{N_R^c}N_R\overline{N_R^c}N_R$ which violates lepton number by 4 units. We note that the first sterile-neutrino-gluon operator arises at dimension seven,

$$\mathcal{O}_{gN}^{(7)} = \overline{N_R^c}N_R\,G^{a,\mu\nu}G_{\mu\nu}^a\,, \qquad (8)$$

where one or both gluon field strengths can be exchanged by their dual.

In the Dirac case, all operators violating lepton number are absent, and in Eq.(5) we need to replace the Majorana kinetic and mass terms by their Dirac fermion counterparts. The operator $\mathcal{O}_{NH}^{(5)}$ is replaced by

$$\mathcal{O}_{NH}^{(5)} = 2\overline{N}NH^\dagger H\,, \qquad (9)$$

as will be explained with the direct detection constraints. Apart from that, we consider the same D6 operators coupling only to the right-handed component of the sterile neutrino, although in principle interactions also with the left-handed components could be present in this case. Other differences arise when annihilation cross sections are important.

## 2.2 Constraints

We need to gather the existing constraints for our $\nu$DMEFT, assuming that $N$ is the only field odd under a $\mathbb{Z}_2$ symmetry. In this section we will use the pure EFT approach and derive limits on the Wilson coefficients and cut-off scale when $N$ couples to either $\tau_R$, $t_R$, or $b_R$. For each of these three cases we consider the operator $\mathcal{O}_{N\tau(t,b)} \equiv \mathcal{O}_{Ne(u,d)}^{33}$ with Wilson coefficient $C_{N\tau(t,b)}$. Relevant constraints come from the relic abundance, as well as direct and indirect detection. The results are shown in Figure 1 for the Majorana and Dirac cases. These three scenarios have been chosen for their simplicity and are not the only viable EFT scenarios. In particular, combinations of several operators may be present with Wilson coefficients of similar magnitude. Due to the relaxation of the relic density constraint towards larger $\Lambda$ in the case of several annihilation channels during freeze-out, these scenarios can be less constrained. We discuss one example of such a multi-operator EFT along with the U(1)$_{(B-L)_3}$ model in Section 3.1.

**Relic abundance**  For a valid DM candidate we need to ensure that $\Omega_N h^2 \leq \Omega_{\text{DM}} h^2 = 0.12$. The key ingredient is the annihilation process

$$NN \to \text{SM } \overline{\text{SM}}, \tag{10}$$

where the relevant particles in the final state are defined by the mediator or the corresponding effective operator. In all our cases the annihilation goes into fermion-antifermion pairs. The EFT framework with $\Lambda \gg m_N$ naturally matches the non-relativistic freeze-out scenario, provided the $2 \to 2$ annihilation rate is large enough to predict the observed relic density. We check the correct relic density using MICROMEGAS [50–53]. The black lines in Figure 1 correspond to $\Omega_N h^2 = 0.12$, while the gray area overshoots the observed DM density. Note the different position of the DM-line for Dirac and Majorana fermions, reflecting the different annihilation cross sections. When kinematically allowed, we find the cross sections for the operators $\mathcal{O}_{Nf}$ with a right-handed singlet $f = \tau, t, b$ to read

$$\langle \sigma v_{\text{rel}} \rangle_{Nf}^{\text{M}} = \frac{|C_{Nf}|^2}{8\pi\Lambda^4} N_{\text{c}} \sqrt{1 - \frac{m_f^2}{m_N^2}} \left( m_f^2 + \frac{16m_N^4 - 23m_N^2 m_f^2 + 10m_f^4}{24(m_N^2 - m_f^2)} v_{\text{rel}}^2 \right), \tag{11}$$

for Majorana DM and up to first order in average squared relative DM velocity $v_{\text{rel}}^2$ by direct computation. For Dirac DM the annihilation rate is

$$\langle \sigma v_{\text{rel}} \rangle_{Nf}^{\text{D}} = \frac{|C_{Nf}|^2}{16\pi\Lambda^4} N_{\text{c}} \, m_N^2 \sqrt{1 - \frac{m_f^2}{m_N^2}}. \tag{12}$$

Here we include the color factor $N_{\text{c}} = 1$ for $f = \tau$ and $N_{\text{c}} = 3$ for $f = t, b$. In the Dirac case, we drop terms of order $v_{\text{rel}}^2$. This should not be done in the Majorana case, since the term of order $v_{\text{rel}}^0$ is proportional to $m_f^2$ instead of $m_N^2$. The next-to-leading order $v_{\text{rel}}^2$ becomes relevant either during freeze-out, when $v_{\text{rel}} \sim \sqrt{T_f / m_f} \sim 10^{-1}$ for a WIMP-like scenario, or even in the indirect detection of DM annihilations today ($v_{\text{rel}} \sim 10^{-3}$), if $m_N \gg m_f$. For operators $\mathcal{O}_{Nf}$ with a doublet representation $f$, one simply adds the cross sections for $t$ and $b$ or $v_\tau$ and $\tau$

Table 2: Mixed fermion-boson D6 operators, giving rise to neutrino interactions, including only SM fields (upper) and including SM fields and right-handed sterile neutrino singlets (lower). Operator name conventions adapted from Ref. [48].

| | $\psi^2 H^3$ | | $\psi^2 X H$ | | $\psi^2 H^2$ |
|---|---|---|---|---|---|
| $\mathcal{O}_{eH}$ | $(H^\dagger H)(\bar{l}_\alpha e_\beta H)$ | $\mathcal{O}_{eW}$ | $(\bar{l}_\alpha \sigma^{\mu\nu} e_\beta)\tau^I H W_{\mu\nu}^I$ | $\mathcal{O}_{Hl}^{(1)}$ | $i\left(H^\dagger \overleftrightarrow{D}_\mu H\right)\left(\bar{l}_\alpha \gamma^\mu l_\beta\right)$ |
| $\mathcal{O}_{uH}$ | $(H^\dagger H)(\bar{q}_\alpha u_\beta \widetilde{H})$ | $\mathcal{O}_{eB}$ | $(\bar{l}_\alpha \sigma^{\mu\nu} e_\beta)H B_{\mu\nu}$ | $\mathcal{O}_{Hl}^{(3)}$ | $i\left(H^\dagger \tau^I \overleftrightarrow{D}_\mu H\right)\left(\bar{l}_\alpha \tau^I \gamma^\mu l_\beta\right)$ |
| $\mathcal{O}_{dH}$ | $(H^\dagger H)(\bar{q}_\alpha d_\beta H)$ | $\mathcal{O}_{uW}$ | $(\bar{q}_\alpha \sigma^{\mu\nu} u_\beta)\tau^I \widetilde{H} W_{\mu\nu}^I$ | $\mathcal{O}_{Hq}^{(1)}$ | $i\left(H^\dagger \overleftrightarrow{D}_\mu H\right)\left(\bar{q}_\alpha \gamma^\mu q_\beta\right)$ |
| | | $\mathcal{O}_{uB}$ | $(\bar{q}_\alpha \sigma^{\mu\nu} u_\beta)\widetilde{H} B_{\mu\nu}$ | $\mathcal{O}_{Hq}^{(3)}$ | $i\left(H^\dagger \tau^I \overleftrightarrow{D}_\mu H\right)\left(\bar{q}_\alpha \tau^I \gamma^\mu q_\beta\right)$ |
| | | $\mathcal{O}_{dW}$ | $(\bar{q}_\alpha \sigma^{\mu\nu} d_\beta)\tau^I H W_{\mu\nu}^I$ | $\mathcal{O}_{He}$ | $i\left(H^\dagger \overleftrightarrow{D}_\mu H\right)\left(\bar{e}_\alpha \gamma^\mu e_\beta\right)$ |
| | | $\mathcal{O}_{dB}$ | $(\bar{q}_\alpha \sigma^{\mu\nu} d_\beta)H B_{\mu\nu}$ | $\mathcal{O}_{Hu}$ | $i\left(H^\dagger \overleftrightarrow{D}_\mu H\right)\left(\bar{u}_\alpha \gamma^\mu u_\beta\right)$ |
| | | | | $\mathcal{O}_{Hd}$ | $i\left(H^\dagger \overleftrightarrow{D}_\mu H\right)\left(\bar{d}_\alpha \gamma^\mu d_\beta\right)$ |
| | | | | $\mathcal{O}_{Hud}$ | $i\left(\widetilde{H}^\dagger D_\mu H\right)\left(\bar{u}_\alpha \gamma^\mu d_\beta\right)$ |
| $\mathcal{O}_{NlH}$ | $(H^\dagger H)(\bar{l}_\alpha N_\beta \widetilde{H})$ | $\mathcal{O}_{NW}$ | $(\bar{l}_\alpha \sigma^{\mu\nu} N_\beta)\tau^I \widetilde{H} W_{\mu\nu}^I$ | $\mathcal{O}_{HN}$ | $i\left(H^\dagger \overleftrightarrow{D}_\mu H\right)\left(\bar{N}_\alpha \gamma^\mu N_\beta\right)$ |
| | | $\mathcal{O}_{NB}$ | $(\bar{l}_\alpha \sigma^{\mu\nu} N_\beta)\widetilde{H} B_{\mu\nu}$ | $\mathcal{O}_{HNe}$ | $i\left(\widetilde{H}^\dagger D_\mu H\right)\left(\bar{N}_\alpha \gamma^\mu e_\beta\right)$ |

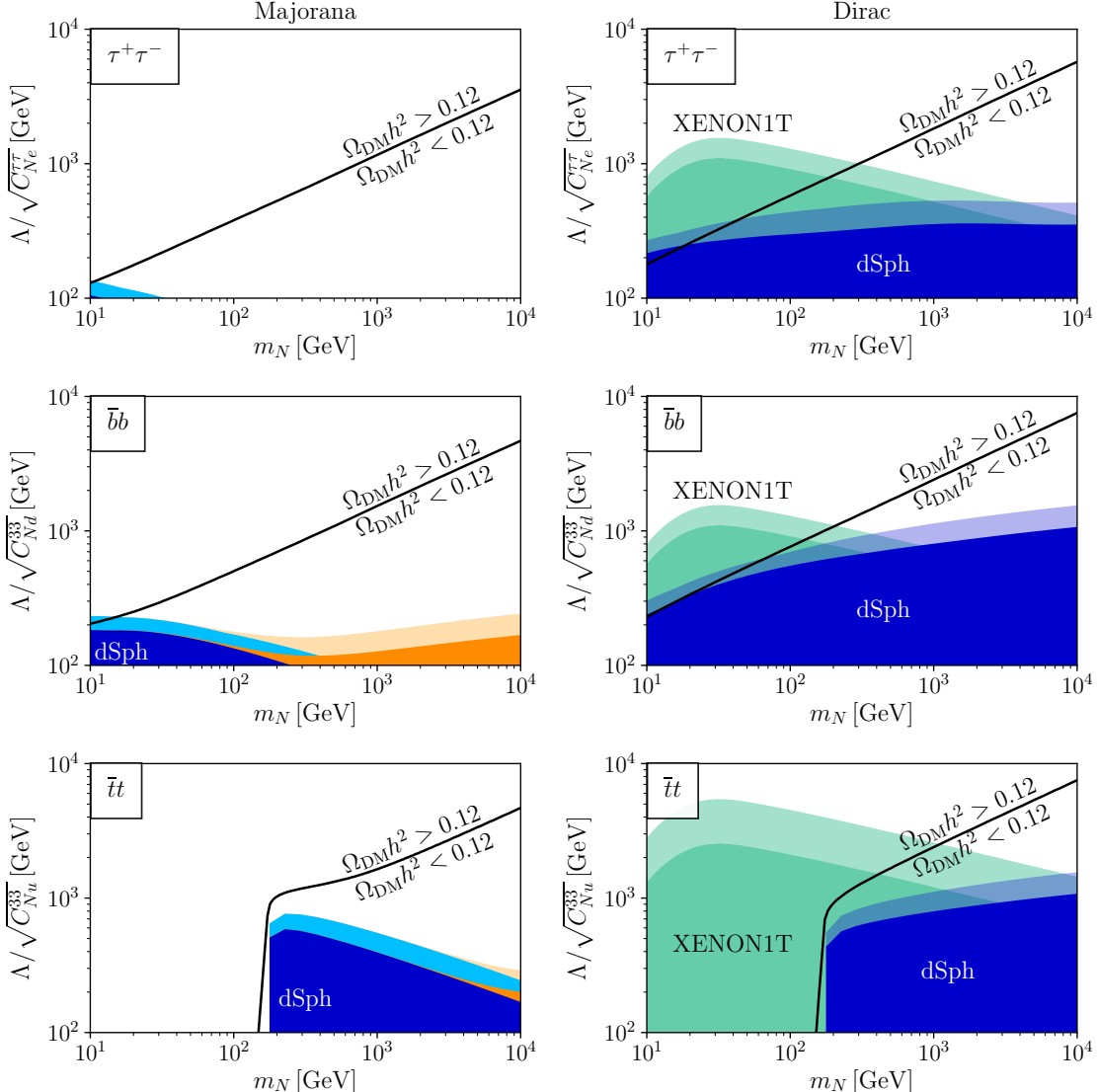

Figure 1: New physics scale required by the relic abundance for the third generation EFT scenarios (black line), assuming Majorana (left) and Dirac (right) sterile neutrinos. Regions excluded by indirect detection from dwarf spheroidals are shown in blue. The light blue region corresponds to the variation of the $J$-factors within their 68% confidence bands. The orange bands show the indirect detection exclusion when we include the loop-induced D7 operator, Eq.(8). Direct detection limits from XENON1T (extrapolated between 1 TeV and 10 TeV) assuming 100 GeV $< \Lambda <$ 10 TeV are shown in teal, the weaker bounds corresponding to $\Lambda = 100$ GeV (200 GeV in the $\bar{t}t$-case). Direct and indirect detection constraints assume $\Omega_{\mathrm{DM}} h^2 = 0.12$ everywhere.

respectively, e.g. in the Dirac case

$$\langle \sigma v_{\mathrm{rel}} \rangle^{\mathrm{D}}_{Nq} = \frac{|C_{Nq}|^2}{16\pi \Lambda^4} N_{\mathrm{c}} \, m_N^2 \left( \sqrt{1 - \frac{m_t^2}{m_N^2}} + \sqrt{1 - \frac{m_b^2}{m_N^2}} \right). \tag{13}$$

For the lepton case with operator $\mathcal{O}_{N\ell}$ we set $N_{\mathrm{c}} = 1$ and replace the masses $m_b$ with $m_\tau$ and $m_t$ with $m_{\nu_\tau} = 0$.

**Direct detection**    The most promising channel to directly detect multi-GeV dark matter such as the traditional WIMP is elastic scattering off nuclei, searched for at experiments such as XENON1T [54]. To contribute to this scattering, the new physics must couple DM to light quarks or gluons at nuclear energy scales. This may happen in various ways at tree level or loop level. Mapping these interactions on non-relativistic nuclear scattering theory allows one to compare the predicted scattering cross section with experimental bounds, see e.g. Ref. [55] for a description of a mapping from the UV theory to the nucleon-level theory.

For our $\nu$DMEFT the scattering off light quarks is described by four-fermion operators $\mathcal{O}_{Nf}$, just as in any WIMP dark matter scenario with an underlying $\mathbb{Z}_2$ symmetry [56]. For this to be detectable, the operators $\mathcal{O}_{Nq}$, $\mathcal{O}_{Nu}$, or $\mathcal{O}_{Nd}$ with quark flavor indices 1 or 2 must be present. However, even if their Wilson coefficients vanish at the weak scale, there will be a non-vanishing coupling at the nuclear scale induced by RG-running. Technically, this means that one needs to map the Wilson coefficients to the appropriate EFT of the SM extended by a right-handed neutrino below the weak scale. This mapping has been discussed in Ref. [35] for neutrino interactions and recently been given completely [39]. We use RUNDM [55] to check whether the running-induced $N$-nucleus coupling is or is not within the reach of current experiments. We find that in the Majorana case, when the $N$-bilinear reduces to an axial current, $\overline{N}\gamma^\mu\gamma^5 N$, only very weakly constrained operators of non-relativistic scattering theory are generated. We typically find couplings of order $10^{-4}$ to $10^{-9}$ for operators 4, 8, and 9 defined in Ref. [57], well below current XENON limits [58].

In the Dirac case, however, the $N$-bilinear involves a vector coupling $1/2\overline{N}\gamma^\mu N$. In this case, the more stringently constrained $\mathcal{O}_1$ in non-relativistic scattering theory,

$$\mathcal{O}_1 = 1_N 1_x\,, \tag{14}$$

where $x$ refers to the nucleon $p$ or $n$, is generated. In the scenarios of $\tau$- and $b$-coupling the proton coupling dominates, while in the scenario of $t$-coupling the neutron coupling dominates. Since this is the operator related to spin-independent scattering, we may use [59]

$$\sigma_{\mathrm{SI}} = \frac{\mu_x^2}{\pi}(\mathcal{C}_x)^2\,, \tag{15}$$

with the reduced nucleon mass $\mu_x = m_N m_x/(m_N + m_x)$, to recast constraints on the WIMP-nucleon cross section from XENON1T as constraints on the Wilson coefficients $\mathcal{C}_x$ of the (relativistic) nucleon-level EFT. We compare these coefficients to the ones generated via RUNDM. In contrast to relic density and indirect detection constraints, these running-induced bounds are dependent not only on $\Lambda/\sqrt{C}$ but on the explicit scale of new physics $\Lambda$, i.e. the mediator mass. For lighter new physics, the constraint is weaker because the running effect is smaller. As soon as we impose the relic density constraint and restrict $\Lambda > 100$ GeV, as suitable for the conditions of our EFT, we find lower bounds on the dark matter mass of

$$m_N \geq \begin{cases} 185 \text{ GeV} & \text{for } \tau^+\tau^-\,, \\ 126 \text{ GeV} & \text{for } \overline{b}b\,, \qquad \text{(Dirac case)} \\ 1002 \text{ GeV} & \text{for } \overline{t}t\,. \end{cases} \tag{16}$$

For illustration of the dependence on the mediator mass scale $\Lambda$, we show in Figure 1 the XENON1T bounds in the range of $100(200)\,\text{GeV} \leq \Lambda \leq 10$ TeV, where the boundary in brackets refers to the $\overline{t}t$ case, assuming $\Omega_{\mathrm{DM}}h^2 = 0.12$ even when not sitting on the black line where this is predicted by freeze-out. Between 1 TeV and 10 TeV we extrapolate the bounds from XENON1T, since this range has not yet been explicitly covered in current publications. Only the $\overline{t}t$ bound is affected mildly by this extrapolation, namely by a shift from 1000 GeV to 1002 GeV.

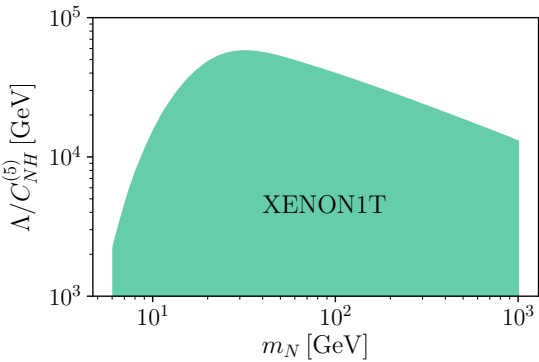

Figure 2: Direct detection constraints from XENON1T [54] on the operator $\mathcal{O}_{NH}^{(5)}$ introduced in Eq.(7). The area is excluded at 90% CL.

Two more operators may contribute to a light-quark coupling, namely $\mathcal{O}_{HN}$ and $\mathcal{O}_{NH}^{(5)}$. The former will not be encountered in this paper. The latter introduces a Higgs portal coupling $NNh$, as mentioned after Eq.(7). This Higgs portal coupling is severely constrained by direct detection. To quantify this, we calculate the spin-independent elastic $N$-nucleus scattering cross section in the zero-momentum-transfer limit [60]

$$\sigma_{\text{SI}} = \frac{4}{\pi} \mu_x^2 f_x^2 \,, \tag{17}$$

where

$$f_x = m_x \frac{C_{NH}^{(5)}}{\Lambda m_h^2} \left( \sum_{q=u,d,s} f_{Tq}^x + \frac{2}{9} f_{TG}^x \right) \tag{18}$$

for $x = p, n$, where we take for $f_{Tq}^x$ and $f_{TG}^x$ the same values as applied in Ref. [61]. We compare the prediction for the Higgs-mediated direct detection cross section with the latest XENON1T results [54] to derive 90% CL lower limits on $\Lambda/C_{NH}^{(5)}$. In the sensitive range of DM masses (6-1000 GeV), we find $\Lambda/C_{NH}^{(5)} > 6 \cdot 10^4$ GeV to be consistent with all masses and $\Lambda/C_{NH}^{(5)} < 2 \cdot 10^3$ GeV to be excluded for all masses. In Figure 2 we show the mass-dependent limit on $\Lambda/C_{NH}^{(5)}$ derived under the assumption that the entire cross section is determined by $\mathcal{O}_{NH}^{(5)}$. The operator $\mathcal{O}_{NH}^{(5)}$ is so strongly constrained, that it is often justified to neglect the annihilation channel $NN \to h$ when considering a freeze-out through four-fermion interactions. Strictly speaking, the Higgs channel may still play a role if the DM mass is close to the resonance $m_N \sim m_h/2$. We do not consider this possibility further in this work. On the other hand, this strong constraint allows to test the models presented in Section 3 in direct detection even when the operator is only generated at loop level. In the EFT scenarios considered in Figure 1, we assume $\Lambda/C_{NH}^{(5)} > 6 \cdot 10^4$ GeV such that there is no restriction in the parameter space from direct detection. In the Dirac case, we included a factor of two in Eq.(9), to keep the direct detection cross section equal to the Majorana case for the same coefficient $C_{NH}^{(5)}$.

In addition to quarks, DM may interact with gluons in the nucleus. This interaction is described by operators like $\mathcal{O}_{gN}^{(7)}$ of at least dimension seven. We ignore it for the D6-Lagrangian considered here. If DM couples to quarks, this operator is necessarily generated via loops. It can be more relevant than the lower-dimensional four-fermion operators, for instance for the Higgs portal, where the light quark Yukawas are tiny and the top loop does not decouple. For other mediators coupling only to third-generation fermions, the power suppression from the loop will weaken the direct detection limits. As an example, for the model described in

Section 3.2 a Higgs portal and a gluon coupling, both generated at one loop, are the relevant contributors to direct detection.

**Indirect detection** The $\nu$DMEFT operators relevant for indirect detection are typically the same as the operators leading to the observed relic density. A significant mismatch between these two processes occurs only when the annihilation process involves a narrow $s$-channel resonance, which cannot happen in the EFT description.

We consider the search for gamma rays from DM annihilation in dwarf spheroidal galaxies (dSphs) by Fermi-LAT [62]. The expected gamma ray flux in the energy range between $E_{\text{min}}$ and $E_{\text{max}}$, and allowing for more than one annihilation process ($j$), is given by

$$\phi = \frac{J}{8\pi m_N^2} \sum_j \langle \sigma v_{\text{rel}} \rangle_j \int_{E_{\text{min}}}^{E_{\text{max}}} \frac{\mathrm{d}N_{\gamma,j}}{\mathrm{d}E_\gamma} \mathrm{d}E_\gamma \,, \tag{19}$$

where $\langle \sigma v_{\text{rel}} \rangle_j$ denote the velocity-averaged annihilation cross sections and $\mathrm{d}N_{\gamma,j}/\mathrm{d}E_\gamma$ the corresponding photon spectra. In addition, this formula includes a $J$-factor for each considered object. The Fermi-LAT and DES collaborations published bin-by-bin likelihoods for the gamma ray spectra of a number of dSphs [62]. We compare the predictions from Eq.(19) with these bin-by-bin likelihoods for each dSph to derive limits on the annihilation rates. The inputs include a set of dSphs, their $J$-factors with corresponding uncertainty, and $\mathrm{d}N_{\gamma,j}/\mathrm{d}E_\gamma$. The calculation of the photon spectra requires particle showers and we employ the spectra provided in Ref. [63].

We include all 19 dSphs with kinematically determined $J$-factors in Table 1 of Ref. [62]. For the $J$-factors we employ the more recent results in Ref. [64] and estimate the uncertainties by the 68% confidence interval given in that reference. For a given $m_N$ we then use the best of the 19 limits on $\langle \sigma v_{\text{rel}} \rangle_j$ using the upper 68% CL bound on the $J$-factors as the best-case limit, and the best out of the 19 limits using the lower 68% CL bound as the worst-case limit. This defines an exclusion band between those two values. We also note that the $J$-factors from Ref. [64] are in most cases slightly lower than previous estimates, so our limits are slightly weakened. The limits obtained by us with this procedure assuming constant $\Omega h^2 = 0.12$ are shown in Figure 1. The light (dark) blue areas correspond to the best-case (worst-case) scenario of the $J$-factors being at the upper (lower) boundary of their 68% CL region. Note the different position of the indirect detection areas for Dirac and Majorana fermions, which is caused by the different form of the annihilation cross sections, see above. From the intersection of the black line with the blue areas, we can derive the following lower limits on the DM mass,

$$m_N \geq \begin{cases} 6 \text{ - } 10 \text{ GeV} & \text{for } \tau^+\tau^-, \\ 6 \text{ - } 14 \text{ GeV} & \text{for } \bar{b}b, \end{cases} \quad \text{(Majorana case)} \tag{20}$$

and

$$m_N \geq \begin{cases} 18 \text{ - } 41 \text{ GeV} & \text{for } \tau^+\tau^-, \\ 11 \text{ - } 57 \text{ GeV} & \text{for } \bar{b}b. \end{cases} \quad \text{(Dirac case)} \tag{21}$$

In the $\bar{t}t$ case, there is no intersection and the whole parameter range is consistent with indirect detection.

We conclude the discussion of indirect detection by commenting on the accuracy of the EFT expansion at dimension six. In principle, there can be higher order operators like the aforementioned D7 operator $\mathcal{O}_{gN}^{(7)}$ with large Wilson coeffecients that influence the indirect detection signal. When we assume a given EFT fit like the single operator scenarios, and are constructing a model that generates the corresponding operators at dimension six, one should check

for potential higher order operators which can spoil the accuracy of the D6 fit. As an example, we refer to existing literature showing that a simple model generating the $\bar{t}t$ and $\bar{b}b$ scenarios features a contribution to the gamma ray spectrum from loop-induced $NN \rightarrow gg$ annihilation [65], which can in some cases be significant. It can be significant only because the Majorana annihilation cross section into fermion pairs is suppressed at present times, since the second term in Eq.(11) involves a factor $v_{\text{rel}}^2 \sim 10^{-6}$. We can apply the results from Refs. [66, 67], which give expressions for the loop-induced annihilation rates of neutralinos into two gluons, mediated by tops and stops or bottoms and sbottoms, respectively. The stop corresponds to the scalar leptoquark of our UV-complete theory in Section 3.2, which generates the EFT $\bar{t}t$ case discussed here. Analogous results for vector leptoquarks, corresponding to the $\bar{b}b$ case as outlined in Section 3.3, are not available in the literature. Following the same procedure as above and setting $v_{\text{rel}} = 10^{-3}$, we include this channel in the derivation of indirect detection constraints. The constraints for a combination of both fermionic and gluonic annihilation channels are seen as the orange continuations in the $\bar{b}b$ and $\bar{t}t$ plots of Figure 1. The effect is less pronounced in the $\bar{t}t$ case, since the first term in in Eq.(11) is not as suppressed for top quarks with their large mass. In both cases the effect is by far not strong enough to affect the lower limit on $m_N$ from indirect searches discussed above, while in the $\bar{\tau}\tau$ case, this channel is expected to be irrelevant due to $\tau$ leptons being color singlets. In conclusion, it turns out that the indirect detection signals of the UV complete models considered in this work are indeed accurately described by the dimension-six expansion discussed in this section.

## 3   $\nu$DMEFT representing models

While in some instances effective theories can safely be viewed as stand-alone theories, we know that for a DMEFT it is crucial that the effective theory can also be justified as a low-energy approximation to classes of UV-complete models [20, 27]. Following this philosophy, we need to compare relevant theory predictions between the DMEFT and corresponding models with propagating mediators. The observables we need to consider are DM annihilation predicting the freeze-out relic density, direct detection, indirect detection, flavor constraints, and collider searches. Some scenarios where the LHC has obvious potential do not provide enthusiastic support for global DMEFT analyses [17, 27]:

- tree-level $s$-channel vector mediator coupling to first-generation quarks: the mediator is strongly constrained by LHC resonance searches. In the allowed parameter range the observed relic density requires an $s$-channel funnel and invalidates the EFT approach;

- loop-level $s$-channel scalar mediator coupling to third-generation quarks at tree level and to gluons at loop level. DM can annihilate into heavy quarks, gluons, and mediator pairs, challenging the EFT picture. Collider searches for resonant mediator production lack sensitivity;

- tree-level $t$-channel scalar mediator coupling to first-generation quarks: the mediator is constrained by LHC pair production. In the remaining parameter range the annihilation rate is too small to explain the observed relic density;

- loop-level $t$-channel mediator coupling to third-generation quarks at tree level and mediating a DM coupling to gluons at loop level. DM can annihilate into heavy quarks and into gluons, challenging a fixed-dimension EFT approach. Nonetheless, we revisit this class and identify suitable regions of the parameter space where the D6-EFT is consistent except for the LHC signatures.

For these classes of mediators the main observation is that mostly the LHC constraints on the mediators make it hard to predict the observed relic density with a DMEFT.

The $\nu$DMEFT adds a new set of UV-models with the promising feature that they include

tree-level mediators which only couple to third-generation fermions. For an EFT approach to make sense, it needs to represent more than one specific UV-model. Examples for such models are a gauge extension only coupling to third-generation fermions in Section 3.1, a scalar third-generation leptoquark coupling to $t$-quarks in Section 3.2 (which represents the loop-level $t$-channel mediator class), and a third-generation vector leptoquark coupling to $b$-quarks in Section 3.3. For these three UV-completions we apply the EFT framework to the relic density, direct and indirect detection. For the LHC we accept that on-shell mediator production cannot be analyzed in the EFT framework [23, 24], so we translate the EFT constraints from cosmological constraints into full-model parameter regions which should be tested at the LHC.

## 3.1 Gauging third-generation $(B-L)$

For gauge extensions it is attractive to start with global symmetry groups which do not develop anomalies when gauged [68, 69]. Because anomalies do not require cancellations between generations we gauge $B-L$ for the third generation only and avoid most constraints. For our purpose of explaining DM, we need one additional SM-singlet scalar $\Phi$, which carries $(B-L)_3$ charge $+2$ and breaks the additional $U(1)$ symmetry [70]. This model as such cannot explain flavor mixing between the third and first two generations, since the necessary mixed Yukawa interactions are forbidden by the differing $(B-L)_3$-charges. However, there are ways to accommodate flavor mixing through extended particle contents. For instance, one may add scalars and vector-like fermions [71], or a scalar with mixed SM and $(B-L)_3$ charges [72]. Models based on $B-L$ are tailor-made for Majorana fermions, so we do not consider the Dirac option here. With a discrete $\mathbb{Z}_2$ symmetry under which only $N_R$ is odd we find the Lagrangian

$$
\mathcal{L} = \mathcal{L}_{\text{SM}} + i\overline{N}_R \gamma^\mu \partial_\mu N_R + g_X \hat{X}_\mu \overline{N_R} \gamma^\mu N_R - g_X \sum_f q_X^f \hat{X}_\mu \overline{f} \gamma^\mu f - \left( \frac{y}{2} \overline{N_R^c} N_R \Phi + \text{h.c.} \right)
$$
$$
+ (D^\mu \Phi)^\dagger \left( D_\mu \Phi \right) + \mu_\Phi^2 \Phi^\dagger \Phi - \lambda_\Phi \left( \Phi^\dagger \Phi \right)^2 - \lambda_{H\Phi} \left( H^\dagger H \right) \left( \Phi^\dagger \Phi \right)
$$
$$
- \frac{1}{4} \hat{X}^{\mu\nu} \hat{X}_{\mu\nu} - \frac{\epsilon}{2} \hat{X}^{\mu\nu} \hat{B}_{\mu\nu}, \tag{22}
$$

where $\mu_H^2$ and $\mu_\Phi^2$ are chosen positive, $f = l_\tau, \tau_R, q_3, t_R, b_R$, and

$$
D_\mu = \partial_\mu + ig \frac{\tau_i}{2} W_\mu^i + ig' \frac{Y}{2} B_\mu + ig_X Y_{(B-L)_3} \hat{X}_\mu. \tag{23}
$$

The last term in the first line of Eq.(22) is the only Yukawa coupling involving $\Phi$. The $\mathbb{Z}_2$ and $(B-L)_3$ symmetries forbid many terms discussed in Section 2.1. We assume for the kinetic mixing parameter that $\epsilon \ll 1$ in order to satisfy stringent experimental constraints, as detailed later in this section. The field $\Phi$ is a SM singlet, hence only kinetic mixing is present.

**Masses and mixing** Below the weak scale and in unitary gauge, the Higgs field and the additional $U(1)$-breaking scalar $\Phi$ can be expressed as

$$
H = \frac{1}{\sqrt{2}} \begin{pmatrix} 0 \\ v + h \end{pmatrix} \qquad \Phi = \frac{1}{\sqrt{2}} (w + \phi), \tag{24}
$$

where $v$ and $w$ denote vacuum expectation values of $H$ and $\Phi$ respectively. We omit the procedure to obtain the physical masses and currents here and refer to the general case discussed e.g. in Ref. [69]. If $w \gg v$ we can expand the VEVs in $\lambda_{H\Phi} \ll 1$,

$$
w = \frac{\mu_\Phi}{\sqrt{\lambda_\Phi}} - \frac{1}{8\sqrt{\lambda_\Phi}\lambda_H} \frac{\mu_H^2}{\mu_\Phi} \lambda_{H\Phi} + \mathcal{O}(\lambda_{H\Phi}^2),
$$
$$
v = \frac{\mu_H}{\sqrt{\lambda_H}} - \frac{1}{8\sqrt{\lambda_H}\lambda_\Phi} \frac{\mu_\Phi^2}{\mu_H} \lambda_{H\Phi} + \mathcal{O}(\lambda_{H\Phi}^2). \tag{25}
$$

We can describe the small mixing of $h$ and $\phi$ with an angle $\theta \in [-\pi/4, \pi/4]$, given by

$$\tan(2\theta) = \frac{\lambda_{H\Phi} v w}{\lambda_\Phi w^2 - \lambda_H v^2} . \tag{26}$$

The new physics mass spectrum is then

$$
\begin{aligned}
m_N &= \frac{y}{\sqrt{2}} w , \\
m_h^2 &= \lambda_H v^2 + \lambda_\Phi w^2 + (\lambda_H v^2 - \lambda_\Phi w^2)\sqrt{1 + \tan^2(2\theta)} \xrightarrow{\lambda_{H\Phi} \to 0} 2\mu_H^2 , \\
m_\phi^2 &= \lambda_H v^2 + \lambda_\Phi w^2 - (\lambda_H v^2 - \lambda_\Phi w^2)\sqrt{1 + \tan^2(2\theta)} \xrightarrow{\lambda_{H\Phi} \to 0} 2\mu_\Phi^2 , \\
\hat{m}_X &= 2 g_X w ,
\end{aligned}
\tag{27}
$$

where $\hat{m}_X$ is the mass of the non-canonically normalized field $\hat{X}$ below the $U(1)$-breaking scale. In the limit $\hat{m}_X \gg v$ the masses of the physical $Z$ and $Z'$ fields in the broken electroweak phase are

$$
\begin{aligned}
m_Z^2 &= m_{Z,0}^2 \left[ 1 - \frac{m_{Z,0}^2}{\hat{m}_X^2} \epsilon^2 s_W^2 \right] , \\
m_{Z'}^2 &= \hat{m}_X^2 \left[ 1 + \epsilon^2 \left( 1 + \frac{m_{Z,0}^2}{\hat{m}_X^2} s_W^2 \right) \right] ,
\end{aligned}
\tag{28}
$$

where $m_{Z,0} = g v / 2 c_W$ and we drop terms of order $\epsilon^4$ and $m_{Z,0}^4 / \hat{m}_X^4$. The limit $m_N \ll m_{Z'}$ then corresponds to $y \ll g_X \lesssim 1$, if the $U(1)_{(B-L)_3}$ is to remain perturbative.

The additional $Z$-$Z'$ mixing induced at the one-loop level is calculable, but in our case it requires a renormalization of $\epsilon$. In practice, the renormalized $\epsilon$ then becomes a free parameter which must be fixed by a measurement and can be constrained from electroweak precision data, depending on $m_{Z'}$ [73].

**Operator matching at the weak scale**    We assume that the symmetry breaking scale of the $(B-L)_3$ is above the weak scale, so in the matching there is no scalar mixing. In Eq.(24) the heavy VEV is now just $w = \mu_\Phi / \sqrt{\lambda_\Phi}$. The scalar and vector masses are $m_\phi = \sqrt{2}\mu_\Phi$ and $\hat{m}_X = \sqrt{2} g_X w$. The interactions of $\phi$, excluding self-interactions, then are

$$\mathcal{L} \supset \left( -\frac{y}{2\sqrt{2}} \overline{N_R^c} N_R \phi + \text{h.c.} \right) + 2 g_X^2 X^\mu X_\mu \phi (\phi + 2w) - \frac{\lambda_{H\Phi}}{2} H^\dagger H \phi (\phi + 2w) . \tag{29}$$

Any lepton-number violating process, in particular the operators in Eq.(7) must involve an insertion of $w$.

We start with the D5 operators in Eq.(7). Since $\nu$ and $N$ do not mix, the lepton number violation in $N$ is not carried over to the active neutrinos and $\mathcal{O}_{\nu\nu}$ does not appear. Next, $\mathcal{O}_{NB}$ is forbidden for a single Majorana neutrino. However, there exists a tree-level contribution to $\mathcal{O}_{NH}^{(5)}$ proportional to $yw \sim m_N$,

$$\frac{C_{NH}^{(5)}}{\Lambda} = -\frac{y}{2\sqrt{2}} \frac{1}{m_\phi^2} \lambda_{H\Phi} w = -\frac{m_N \lambda_{H\Phi}}{2 m_\phi^2} . \tag{30}$$

This operator has been discussed in Section 2.2 to be potentially relevant for direct detection. However, when the mass of $\phi$ is moderately large, this operator is too suppressed to leave a detectable signal, as will be quantified below.

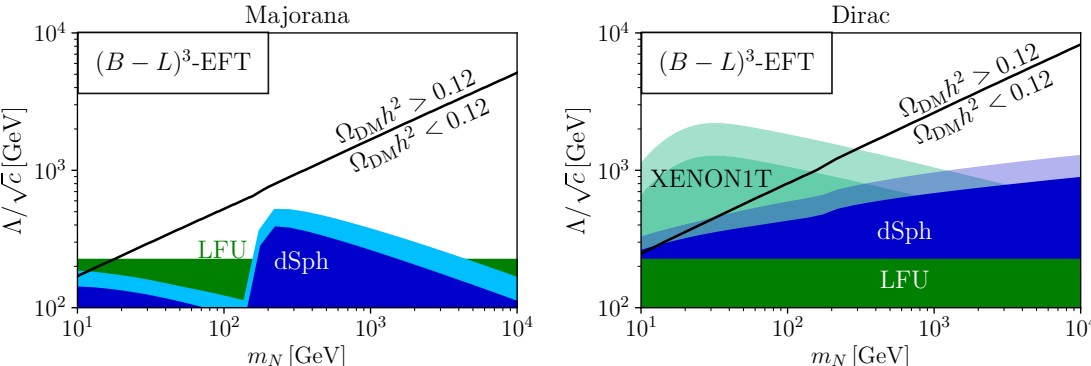

Figure 3: New-physics scale required by the relic abundance for the third generation EFT scenario defined by Eq.(33) (black line) assuming Majorana (left) and Dirac (right) sterile neutrinos. Regions excluded by indirect detection from dwarf spheroidals are shown in blue. The light blue region corresponds to the variation of the $J$-factors within their 68% confidence bands. The bound from lepton flavor universality tests is shown in green. Direct detection limits from XENON1T (extrapolated between 1 TeV and 10 TeV) assuming $100\,\text{GeV} < \Lambda < 10\,\text{TeV}$ are shown in teal, the weaker bounds corresponding to $\Lambda = 100\,\text{GeV}$. Direct and indirect detection constraints assume $\Omega_{\text{DM}}h^2 = 0.12$ everywhere.

The 4-fermion operators $\mathcal{O}_{ff'}$ with $f, f' = l, e, N, q, u, d$ are generated at tree level by integrating out the heavy vector $X$. Depending on the generation indices this requires an additional 0, 1, or 2 powers of $\epsilon$. The tree-level contribution to the corresponding Wilson coefficient reads

$$\frac{C_{ff'}^{\alpha\alpha\beta\beta}}{\Lambda^2} = \frac{g_X^2}{m_X^2}\left[q_X^{f,\alpha}q_X^{f',\beta} + \frac{g'}{g_X}\frac{1}{2}\left(q_X^{f,\alpha}q_Y^{f'} + q_Y^{f}q_X^{f',\beta}\right)\epsilon + \left(q_X^{f,\alpha}q_X^{f',\beta} + \frac{g'^2}{g_X^2}\frac{1}{4}q_Y^{f}q_Y^{f'}\right)\epsilon^2\right], \quad (31)$$

where $q_{X,Y}^f$ are the charges of fermion $f$ under the $U(1)_X$ and hypercharge symmetries. If neither $f_\alpha$ nor $f'_\beta$ carries a $(B-L)_3$ charge, i.e. $\alpha, \beta = 1, 2$, the contribution is suppressed by a factor of $\epsilon^2$. This leads to a natural hierarchy in contributions to the operators. Third-generation components of the operators $\mathcal{O}_{ff'}$ have the largest contribution of new physics of order $1/m_X^2$, whereas generation-mixed operators have contributions of order $\epsilon/m_X^2$, and operators not involving the third generation are suppressed by $\epsilon^2/m_X^2$. The other 4-fermion operators receive no tree-level contribution. The operators $\mathcal{O}_{eNud}$, $\mathcal{O}_{Nlel}$, $\mathcal{O}_{lNqd}$, $\mathcal{O}'_{lNqd}$, and $\mathcal{O}_{lNuq}$ are forbidden by the $\mathbb{Z}_2$ symmetry.

The operators of type $\psi^2 H^3$ in Table 2 are generated via the quartic SM-Higgs coupling. An additional contribution is generated at tree level by an internal $\phi$-line,

$$\lambda_H = \lambda_H^{\text{SM}} - \frac{\lambda_{H\phi}^2}{4}\frac{w^2}{m_\phi^2} = \lambda_H^{\text{SM}} - \frac{\lambda_{H\phi}^2}{8}\frac{1}{\lambda_\Phi}. \quad (32)$$

The operators of type $\psi^2 H^2$ are generated by $B$ and $X$ exchange for singlet bilinears and by $W$ exchange for triplet bilinears like $\mathcal{O}_{Hl}^{(3)}$. Whenever $X$ is the mediator their suppression is at least $\epsilon/m_X^2$, since the $X$-Higgs coupling is generated by kinetic mixing with $B$. The same is true for $\mathcal{O}_{HN}$. Finally, the $L$-violating D6 operator $\mathcal{O}_{N^4}$ is also generated by scalar $\Phi$ exchange.

We will generally assume that the scalar $\Phi$ is heavier than $N$ and that the scalar mixing $\lambda_{H\Phi}$, as well as the $Z$-$Z'$ mixing $\epsilon$ are small. This leaves us, to 0th order in $\lambda_{H\Phi}$ and $\epsilon$, only

with D6 four-fermion operators of the third generation. The following Wilson coefficients are non-zero:

$$C_{LL} = g_X^2 \equiv c\,, \qquad C_{lq}^{(1)} = C_{LQ} = -\frac{g_X^2}{3} \equiv -\frac{c}{3}\,, \qquad C_{qq}^{(1)} = C_{QQ} = \frac{g_X^2}{9} \equiv \frac{c}{9}\,, \qquad (33)$$

where $LL = ll, NN, ee, Nl, le, Ne$, $LQ = lu, ld, Nq, Nu, Nd, qe, eu, ed$, and $QQ = uu, dd, qu, qd,$ $ud$, and flavor indices 3 are understood. The EFT scale is given by $\Lambda = m_{Z'}$. We will further consider $C_{NH}^{(5)}$ only in the context of direct detection in order to constrain the scalar mixing. For these operators we need to discuss the relevant constraints at EFT level and specific to a given UV-completion.

**Relic density (EFT)**   We implement the model via FeynRules [74] and calculate the DM relic density after freeze-out using MICROMEGAS [50–53]. In analogy to the single operator scenarios shown in Figure 1, we also calculate the relic density for an EFT defined by the Wilson coefficients in Eq.(33). The result in terms of the mass parameter $\Lambda/\sqrt{c}$ is shown in Figure 3 along with constraints from indirect detection and lepton flavor universality to be discussed below.

**Direct detection (EFT)**   Looking at direct detection constraints in the Majorana case, the D6 operators discussed above are extremely poorly constrained, again, since the running-induced nucleon couplings are small. Direct detection, however, directly probes $\mathcal{O}_{NH}^{(5)}$, which is determined by the combination of heavy scalar mass and scalar mixing, $m_\phi/\sqrt{\lambda_{H\Phi}}$. In Figure 4 we show the XENON1T limits [54]. The advantage of the EFT approach is that we may immediately recast the bounds on the Wilson coefficient shown in Figure 2 onto bounds on the scalar mass and mixing using Eq.(30). In the Dirac case, when the dark matter couples to the SM fermions through a vector current in addition to the axial current, the RG-running enhances the spin-independent scattering cross section, as discussed in Section 2.2. Upon rescaling for the correct relic abundance, this leads to a lower bound on the dark matter mass of 146 GeV. Compared to the single-operator scenarios, (16), we find the direct detection constraint to be relieved with respect to the $\tau^+\tau^-$ and $\bar{t}t$ case, but tightened with respect to the $\bar{b}b$ case.

**Indirect detection (EFT)**   We discussed our method for evaluating indirect detection constraints arising from the four-fermion operators in Section 2.2. The result of this evaluation is shown in Figure 3 in blue. The curves are clearly dominated by the rescaled sums of the $\bar{t}t$ and $\bar{b}b$ curves in Figure 1, which provide stronger limits than the $\tau^+\tau^-$ annihilation. In the Majorana case, the blue indirect detection limit leads to a lower limit on the DM mass in the range of $m_N \gtrsim 7\text{-}11$ GeV, while in the Dirac case the limit is $m_N \gtrsim 11\text{-}29$ GeV. In this multiple-operator scenario, we observe that even though the relic density constraint can be satisfied for lower individual annihilation cross-sections, i.e. larger $\Lambda/\sqrt{c}$, indirect detection bounds on the dark matter mass are comparable to those of the single-operator cases summarized in Eq.(20) and (21), since all channels except $\bar{\nu}_\tau \nu_\tau$ that contribute to the relic abundance, also contribute to the gamma ray spectrum.

**Lepton universality (EFT)**   Since in this EFT new interactions among four SM fermions arise only for the third generation, flavor observables can be used to constrain the parameter space. In this case we face constraints from lepton flavor universality tests. In particular, the operators $\mathcal{O}_{lq}, \mathcal{O}_{ld}, \mathcal{O}_{qe}, \mathcal{O}_{ed}$ coupling only to the third generation will affect $b$-meson branching ratios. For the $\Upsilon(1S)$ decay into leptonic final states we use the formula [75,76]

$$\Gamma_{\Upsilon(1S)\to\ell\ell} = 4\alpha^2 Q_b^2 \frac{|R_n(0)|^2}{m_\Upsilon^2} K_\ell\,, \qquad (34)$$

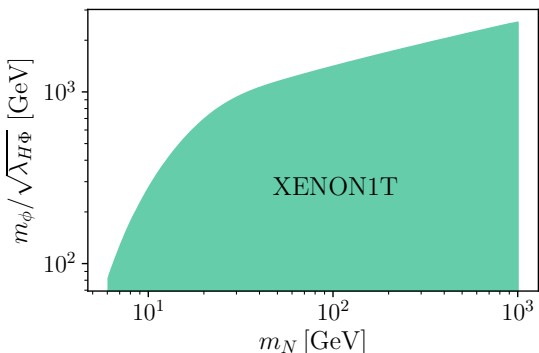

Figure 4: Direct detection constraints from XENON1T [54] on the mass scale $m_\phi/\sqrt{\lambda_{H\Phi}}$ which controls the Wilson coefficient of the operator $\mathcal{O}_{NH}^{(5)}$. The bounds have been obtained applying Eq.(30) to the EFT results shown in Figure 2.

where $\alpha$ denotes the fine-structure constant, $Q_b = -1/3$ the $b$-charge, $R_n(0)$ the non-relativistic radial wave function at the origin, and

$$K_\ell = \left(1 + 2\frac{m_\ell^2}{m_\Upsilon^2}\right)\sqrt{1 - 4\frac{m_\ell^2}{m_\Upsilon^2}} \tag{35}$$

contains the kinematics. The only appearance of the lepton mass is in $K_\ell$. This is why the SM expectation for the ratio $R_{\ell\ell'}$ of decay widths is simply

$$R_{\ell\ell'} = \frac{\Gamma_{\Upsilon(1S)\to\ell\ell}}{\Gamma_{\Upsilon(1S)\to\ell'\ell'}} = \frac{K_\ell}{K_{\ell'}}. \tag{36}$$

Focusing on the third generation, the SM-prediction $R_{\tau\mu} = 0.992$ is consistent with the BaBar measurement $R_{\tau\mu} = 1.005 \pm 0.013(\text{stat.}) \pm 0.022(\text{syst.})$ [76]. Adding statistical and systematic errors quadratically, we use $R_{\tau\mu} = 1.005 \pm 0.026$ to constrain the $\nu$DMEFT.

In this EFT the operators $\mathcal{O}_{lq}$, $\mathcal{O}_{ld}$, $\mathcal{O}_{qe}$, $\mathcal{O}_{ed}$ inducing 4-point interactions between two $b$ and two $\tau$ all have the same Wilson coefficients and therefore add to a vector-like interaction

$$\mathcal{L}_{\text{LFV}(b\tau)} = -\frac{c}{3\Lambda^2}\overline{b}\gamma_\mu b\,\overline{\tau}\gamma^\mu\tau. \tag{37}$$

Their contribution to $\Upsilon$ decays adds directly to the photon-mediated contribution and modifies the branching ratio to

$$\Gamma_{\Upsilon(1S)\to\tau\tau} = 4\alpha^2 Q_b^2 \frac{|R_n(0)|^2}{m_\Upsilon^2} K_\tau \left(1 - \frac{m_\Upsilon^2}{4\pi\alpha Q_b}\frac{c}{3\Lambda^2}\right)^2. \tag{38}$$

Since the effective operator is limited to third-generation fermions it predicts

$$R_{\tau\mu} = \frac{K_\tau}{K_\mu}\left(1 + \frac{m_\Upsilon^2}{4\pi\alpha}\frac{c}{\Lambda^2}\right)^2. \tag{39}$$

We can derive 68% CL limits from the above-mentioned BaBar limit and find $\Lambda/\sqrt{c} \geq 224$ GeV since $c = g_X^2 > 0$. We show the limit as the green area in Figure 3. From the intersection of the green boundary with the black relic density line, we deduce a lower limit on the DM mass consistent with LFU constraints in the EFT regime. In the Majorana case, this yields a stronger lower bound on the DM mass than indirect detection, namely $m_N \gtrsim 17$ GeV, however, at lower CL.

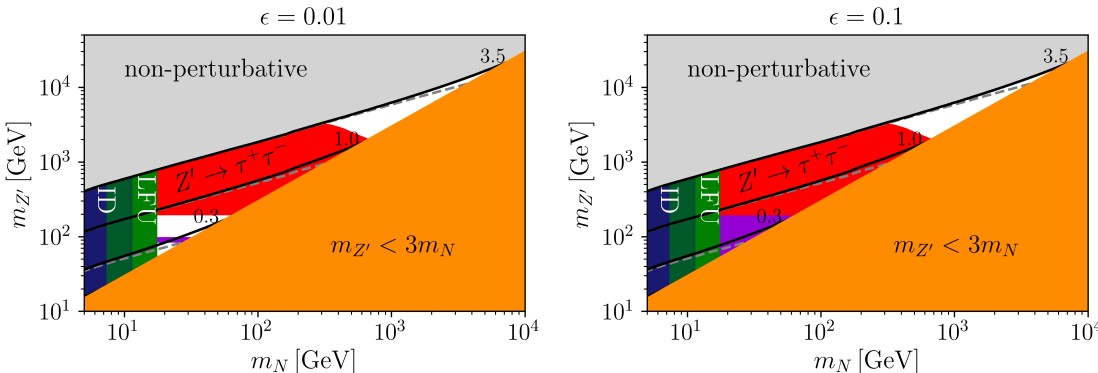

Figure 5: Parameter range of the gauged $(B-L)_3$ model for $Z$-$Z'$ mixing parameters $\epsilon = 0.01$ (left) and $\epsilon = 0.1$ (right). Contours of $g_X$ producing the correct relic abundance are shown in black, while the dashed lines represent the EFT limit upon identifying $\Lambda = m_{Z'}$ and $c = g_X^2$ with $c$ defined in Eq.(33). For a given value of $\epsilon$ the purple areas are ruled out at 95% CL. The area with $m_{Z'} < 3m_N$ is drawn to signify where the EFT language is valid, namely in the upper left corner. Above $g \simeq \sqrt{4\pi} \approx 3.5$ the theory ceases to be perturbative.

**LHC and other constraints on the UV-model**  Turning to the full model, we plot in Figure 5 the $m_N$-$m_{Z'}$ plane and fix at each point the gauge coupling $g_X$ such that we predict the observed relic density. This way we reduce the parameter space $(m_N, m_{Z'}, g_X)$ to two dimensions. We compare this in the same plot to the EFT limit by reducing also the EFT parameter space $(m_N, \Lambda, c)$ to the $m_N$-$m_{Z'}$ plane by identifying $\Lambda = m_{Z'}$ and fixing $c$ through the relic density. For the validity of the EFT approach we require $m_{Z'} < 3m_N$, to ensure that annihilation cannot proceed through an intermediate Breit-Wigner propagator. We note that this definition of the validity of the EFT approach is process-dependent and makes the minimal assumption that a propagating mediator does not contribute for instance to DM annihilation [77].

Using $\sqrt{c} = g_X$ from Eq.(33), we plot contours of fixed $g_X$ as solid lines and contours of fixed $\sqrt{c}$ of the same values 0.3, 1.0, and 3.5 as dashed lines to illustrate the slight deviations of the EFT limit from the exact model. We find that as long as $m_{Z'} > 3m_N$, the freeze-out is nicely described by the effective four-fermion operators. Since for $m_f \ll m_N$ and $v_{\rm rel} \sim 0.1$ the annihilation cross section $\langle \sigma v_{\rm rel} \rangle^{\rm M}_{Nf}$ discussed in Section 2.2 around Eq.(10) is insensitive to $m_f$, the relative importance of the annihilation channels is simply

$$\langle \sigma v_{\rm rel} \rangle_{\tau^+\tau^-} \approx 2 \langle \sigma v_{\rm rel} \rangle_{\bar{\nu}_\tau \nu_\tau} \approx 3 \langle \sigma v_{\rm rel} \rangle_{\bar{b}b} \approx 3 \langle \sigma v_{\rm rel} \rangle_{\bar{t}t} \tag{40}$$

if $m_N \gg m_t$, and without the last approximate equality when $m_b \ll m_N \ll m_t$. Here, the factor of 2 is due to only left-handed tau neutrinos existing, and the factors of 3 are simply a result of the color factors and $(B-L)_3$-charges. Around $m_{Z'} \sim 2m_N$, since $Z'$ is an $s$-channel mediator, $NN \to Z'$ becomes a relevant and immediately resonant annihilation channel during freeze-out. Therefore, we exclude the region $m_{Z'} < 3m_N$ in our plot, since we investigate only the regions described by the EFT. The reader interested in this region is referred to Ref. [70]. We now explain which generic constraints in the EFT picture discussed above and which model-specific constraints give rise to the excluded regions in Figure 5.

1. Perturbativity: at $g_X > \sqrt{4\pi} \approx 3.5$, the theory becomes non-perturbative. Hence our simple tree-level matching becomes unreliable. While this is not strictly excluded, the EFT representing fundamental models turns into a collection of operators and our interpretation ceases to be useful in this regime.

2. Collider production: for a significant portion of the parameter space, $m_{Z'}$ is small enough to be produced on-shell at the LHC. Therefore, the ATLAS and CMS searches for spin-1 resonances produce relevant constraints. It turns out that the channel $Z' \to \tau^+\tau^-$ is the most sensitive one for models such as $(B-L)_3$, where the DM mediator couples predominantly to third generation SM fermions [70, 78]. Currently, ATLAS provides the most stringent limits on the $pp \to Z' \to \tau^+\tau^-$ cross section [79]. Details on the calculation of this contour are delegated to Appendix A.

3. Direct detection: the discussion is carried over from the EFT case: The running-induced coupling of DM to light quarks is too small to be relevant, such that the combination of scalar mass and scalar mixing $m_\phi/\sqrt{\lambda_{H\Phi}}$ is likely the strongest source of a direct detection signal and can be constrained by XENON1T as shown in Figure 4. Assuming this to be satisfied, there is no exclusion from direct detection shown in Figure 5.

4. Indirect detection: The discussion on fermionic annihilations is carried over from the EFT case: As can be seen from Figure 3, in the Majorana case they lead to a lower limit on the DM mass in the range of $m_N \geq 7-11$ GeV varying along the 68% CL region of the $J$-factors. As noted in Section 2.2, it should be ensured in particular in the Majorana case that no D7 operators or loop-induced annihilation channels are substantial compared to the four-fermion operators. Indeed, the processes $NN \to \gamma\gamma, gg$ are possible via a triangle loop diagram. However, since the SM fermions running in the loop have only vector-like couplings to the $Z'$, these processes vanish courtesy of Furry's theorem.

5. Lepton flavor universality: from the pure EFT case, see Eq.(39), a lower bound of $m_N \gtrsim 17$ GeV for combinations of operators can be carried over, which is stronger than the lower bound from indirect detection.

6. $Z$-$Z'$-mixing: as noted earlier, the $Z'$ mixing $\epsilon$ is a free parameter that must be determined by experiment. Any heavy $Z'$ that mixes with the $Z$ boson can be constrained by its effect on $Z$ observables. Along with Ref. [70], we employ the 95% CL bound on $\epsilon$ as a function of $m_{Z'}$ obtained in Ref. [73]. To illustrate the influence of these constraints, we show in Figure 5 two plots, one with $\epsilon = 0.01$ and one with $\epsilon = 0.1$. As can be seen from the purple region, for $\epsilon = 0.01$ only a narrow band around $m_{Z'} \sim m_Z$ is ruled out, leaving open parts of the parameter space below $m_{Z'} < 200$ GeV. The fact that for $\epsilon = 0.1$ masses $m_{Z'} \lesssim 320$ GeV are ruled out completely closes this window for larger mixings. For completeness, we note that when $\epsilon \lesssim 0.005$ all masses $m_{Z'}$ are allowed.

Because this will become relevant later, we briefly comment on the hypothetical case where the $Z'$ only couples to bottom or top quarks. We still obtain limits, for instance, by taking advantage of ATLAS searches for resonant associated scalar production with a subsequent decay to the same quarks, $pp \to q\bar{q}q\bar{q}$ where $q = b, t$ [80, 81]. Details on this evaluation are delegated to Appendix B. We find a minuscule constraint in the top case, as shown later in Figure 6.

## 3.2 Scalar leptoquarks

For a second UV-completion we turn to leptoquarks [82], when the leptoquark is a scalar or a vector. Leptoquarks have recently re-surfaced to explain flavor anomalies [83,84]. The general Lagrangian including all possible couplings between two fermions (SM and sterile neutrino) and one scalar or vector leptoquark was given in Ref. [35], generalizing the list from Ref. [85].

Table 3: Leptoquarks that can couple to SM particles and right-handed neutrino singlets, together with the operators of Table 1 they can generate [35]. Our convention is $Q = I_3 + Y/2$. The cases considered here are $S_1''$ and $U_1''$, generating only effective interactions of our DM fermion $N$ with right-handed up- or down-quarks.

|  | $F$ | Spin | $SU(3)_C$ | $SU(2)_L$ | $U(1)_Y$ | D6-operators |
|---|---|---|---|---|---|---|
| $S_1$ | $-2$ | 0 | $\overline{3}$ | 1 | 2/3 | $\mathcal{O}_{lq}^{(1)}, \mathcal{O}_{Nd}, \mathcal{O}_{lNqd}, \mathcal{O}_{lNqd}',$ $\mathcal{O}_{eluq}, \mathcal{O}_{eNud}$ |
| $S_1'$ | $-2$ | 0 | $\overline{3}$ | 1 | 8/3 |  |
| $S_1''$ | $-2$ | 0 | $\overline{3}$ | 1 | $-4/3$ | $\mathcal{O}_{Nu}$ |
| $S_3$ | $-2$ | 0 | $\overline{3}$ | 3 | 2/3 | $\mathcal{O}_{lq}^{(3)}$ |
| $V_2$ | $-2$ | 1 | $\overline{3}$ | 2 | 5/3 | $\mathcal{O}_{ld}, \mathcal{O}_{elqd}$ |
| $V_2'$ | $-2$ | 1 | $\overline{3}$ | 2 | $-1/3$ | $\mathcal{O}_{Nq}, \mathcal{O}_{lu}, \mathcal{O}_{lNuq}$ |
| $R_2$ | 0 | 0 | 3 | 2 | 7/3 | $\mathcal{O}_{lu}, \mathcal{O}_{eluq}$ |
| $R_2'$ | 0 | 0 | 3 | 2 | 1/3 | $\mathcal{O}_{ld}, \mathcal{O}_{Nq}, \mathcal{O}_{lNqd}, \mathcal{O}_{lNqd}'$ |
| $U_1$ | 0 | 1 | 3 | 1 | 4/3 | $\mathcal{O}_{lq}^{(1)}, \mathcal{O}_{Nu}, \mathcal{O}_{elqd}, \mathcal{O}_{lNuq}, \mathcal{O}_{eNud}$ |
| $U_1'$ | 0 | 1 | 3 | 1 | 10/3 |  |
| $U_1''$ | 0 | 1 | 3 | 1 | $-2/3$ | $\mathcal{O}_{Nd}$ |
| $U_3$ | 0 | 1 | 3 | 3 | 4/3 | $\mathcal{O}_{lq}^{(3)}$ |

We can distinguish leptoquarks with fermion number $F = 3B + L = 0$ and $F = 2$:

$$\begin{aligned}
\mathcal{L}_{F=2} = &\left( s_{1L} \overline{q^c} i\tau_2 l + s_{1e} \overline{u_R^c} e_R + s_{1N} \overline{d_R^c} N \right) S_1 \\
&+ s_1' \overline{d_R^c} e\, S_1' + s_1'' \overline{u_R^c} N\, S_1'' + s_3 \overline{q^c} i\tau_2 \vec{\tau} l\, \vec{S}_3 \\
&+ \left( v_{2R} \overline{q^c}^i \gamma_\mu e_R + v_{2L} \overline{d_R^c} \gamma_\mu l^i \right) \epsilon_{ij} V_2^{\mu,j} \\
&+ \left( v_{2R}' \overline{q^c}^i \gamma_\mu N + v_{2L}' \overline{u_R^c} \gamma_\mu l^i \right) \epsilon_{ij} V_2^{\mu,j\,'} + \text{h.c.},
\end{aligned}$$
(41)

$$\begin{aligned}
\mathcal{L}_{F=0} = &\left( r_{2R} \overline{q}^j e_R + r_{2L} \overline{u_R} l^i \epsilon_{ij} \right) R_2^j \\
&+ \left( r_{2L}' \overline{d_R} l^i \epsilon_{ij} + r_{2R}' \overline{q}^j N \right) R_2^{j\,'} \\
&+ \left( u_{1L} \overline{q} \gamma_\mu l + u_{1de} \overline{d_R} \gamma_\mu e_R + u_{1uN} \overline{u_R} \gamma_\mu N \right) U_1^\mu \\
&+ u_1' \overline{u_R} \gamma_\mu e_R\, U_1^{\mu\,'} + u_1'' \overline{d_R} \gamma_\mu N\, U_1^{\mu\,''} + u_3 \overline{q} \vec{\tau} \gamma_\mu l\, \vec{U}_3^\mu + \text{h.c.}
\end{aligned}$$
(42)

Here $S_1''$ and $U_1''$ only couple to the SM if there are sterile neutrinos. We will focus on those two possibilities. We list the quantum numbers of the leptoquarks in Table 3, along with the $\nu$SMEFT operators that they induce upon integrating out a virtual leptoquark [35]. We will assume here baryon number violation, which rules out the operators

$$\begin{aligned}
\mathcal{L}_{F=2}^{\Delta B} = &s_{1B} l\overline{q} i\tau_2 q^c S_1 + s_{1B}' \overline{u} u^c S_1' + s_{1B}'' \overline{d} d^c S_1'' \\
&+ s_{3B} \overline{q} \vec{\tau} i\tau_2 q^c \vec{S}_3 + s_{2B} \overline{q} \gamma_\mu u^c V_2^\mu + s_{2B}' \overline{q} \gamma_\mu d^c V_2^{\mu\,'},
\end{aligned}$$
(43)

which, together with Eq.(41) may lead to proton decay. Usually, baryon number conservation is assumed to avoid these strong bounds. Finally, we choose to ignore Higgs-portal-like terms of the form $XX^\dagger HH^\dagger$, where $X$ is any of the leptoquarks.

In a first step we focus on the scalar leptoquark $S \equiv S_1'' \sim (\bar{3}, 1, -4/3)$. It is rather secluded from the SM since it has only one interaction term with a SM fermion and sterile neutrinos. It couples only to right-handed up-type quarks, in our case this is the top. Since $S$ then has top-like quantum numbers, it has the features of a top squark in supersymmetry [86]. The DM phenomenology of this framework has been studied in detail in Ref. [87]. It also corresponds to the model considered in Ref. [65], if we fix the involved SM fermions to top quarks.

The Lagrangian relevant for $S$ reads

$$\mathcal{L}_{\mathrm{LQ}} = -m_S^2 S^\dagger S + (D^\mu S)^\dagger D_\mu S + x_t \, \overline{t_R^c} N S + x_t^* (S)^\dagger \overline{N} t_R^c, \tag{44}$$

where

$$D_\mu = \partial_\mu + i g_s T^a G_\mu^a. \tag{45}$$

Besides the coupling to the right-handed up-type quarks, the coupling to gluons is entirely determined by the strong gauge coupling $g_s$ [88]. We need to assume that $S$ is odd under the stabilizing $\mathbb{Z}_2$, because otherwise the Yukawa term in Eq.(44) is not allowed. This type of interaction is possible both for Majorana and Dirac sterile neutrinos and in both cases only the right-handed components will interact. While the coupling of $S$ to the up-type quarks could be fixed to $(0, 0, x_t)$ in flavor space at the new physics scale, there would be small contributions to up and charm quark couplings $x_u$ and $x_c$ from RG running. These are, however, at most of the order $10^{-7} x_t$ and can be neglected for our purposes, since we are only considering the EFT region with significant mass splitting between $S$ and $N$ [87].

**Operator matching**    Straightforwardly, for processes at momentum $p$, for $m_S \gg p$ the effective point interaction after integrating out $S$ reads

$$\mathcal{L}_{\mathrm{eff}} = -\frac{|x_t|^2}{m_S^2} (\overline{t_R^c} N)(\overline{N} t_R^c). \tag{46}$$

By Fierz, this can be rephrased as the operator $\mathcal{O}_{Nt} \equiv C_{Nu}^{33}$ with

$$C_{Nt} = -\frac{|x_t|^2}{2}, \tag{47}$$

when we identify $\Lambda = m_S$. Hence, the effective interaction mimics a vector-mediated $\bar{t} t \leftrightarrow \overline{N} N$ interaction and we can compare the scenario to a $Z'$ coupling only to $t_R$ and $N_R$ inducing the same operator. At one-loop level, both $\mathcal{O}_{NH}^{(5)}$ and $\mathcal{O}_{gN}^{(7)}$ are generated. We only state here the formula for the effective Higgs coupling calculated in Ref. [89] for the Majorana case in the terminology of Ref. [87] (the relation reads $g_{h\chi\chi}/v = 2C_{NH}^{(5)}/\Lambda$) and in the limit of $\sqrt{s}, m_N \ll m_S$,

$$\frac{C_{NH}^{(5)}}{\Lambda} = -\frac{y_t^2 |x_t|^2 m_N}{64 \pi^2 m_t^2} \left( F(r) + \frac{s}{m_t^2} G(r) + \frac{m_N^2}{m_t^2} H(r) \right), \tag{48}$$

where $y_t$ is the SM top Yukawa coupling, $r = m_S^2/m_t^2$ and the functions $F(r)$, $G(r)$, and $H(r)$ are given in Eq.(10) of Ref. [87]. This way the effective Higgs portal coupling is not an independent parameter, but determined by $m_N$, $m_t$, and $m_S$ setting the freeze-out conditions. We omit the explicit Wilson coefficient of $\mathcal{O}_{gN}^{(7)}$, since it is only relevant for indirect detection in our EFT region and in that case we may use the explicit model formulae from Refs. [66, 67]. Let us discuss the constraints arising in this scenario, and compare to the EFT discussion in Section 2.2.

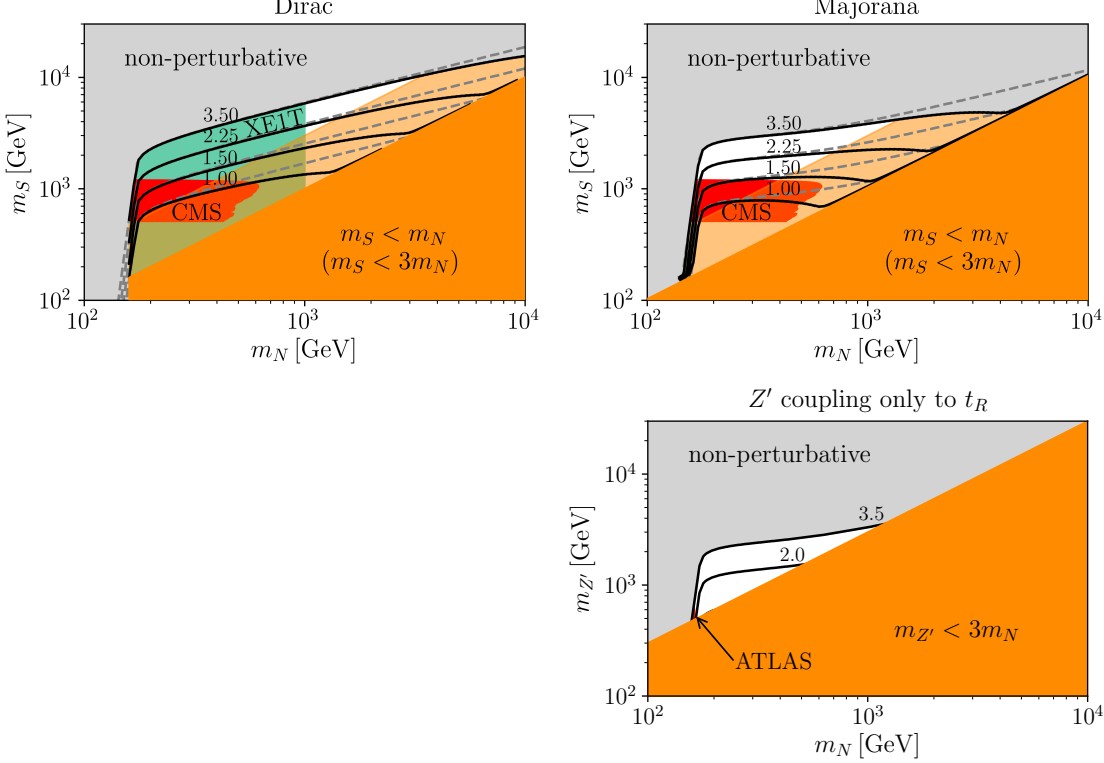

Figure 6: Upper: contours of $|x_t|$ in the $m_N$-$m_S$-plane producing the correct relic abundance. Dashed lines correspond to the EFT limit of the given coupling. Constraints from indirect and direct detection are very weak for Majorana neutrinos. In the case of Dirac neutrinos, direct detection limits from XENON1T are shown in teal. CMS limits are shown in red, perturbativity of the coupling and validity of the EFT description ($m_S < 3m_N$ in orange, $m_S < m_N$ in light orange) are also indicated. Lower: same plot for $Z'$ models coupling only to the $B-L$ charge of right-handed top quarks. Contours of $g_X$ producing the correct relic abundance are shown in black, the $Z$-$Z'$ mixing parameter $\epsilon$ is zero. Note the heroic ATLAS blob in the lower plot.

**Relic density** We implement the model via FeynRules [74] and calculate the DM relic density after freeze-out using MICROMEGAS [50–53]. In Figure 6 we plot the $m_N$-$m_S$ plane and fix at each point $|x_t|$ such that the observed DM relic density is obtained. We compare this to the EFT limit of the model described by the matching Eq.(47) together with $\Lambda = m_S$. This means that for each contour of constant $|x_t|$ we associate a dashed EFT contour with constant $\sqrt{|C_{Nt}|} = |x_t|/\sqrt{2}$, where the $m_S$ coordinates are given by

$$m_S(m_N) = \frac{\Lambda}{\sqrt{|C_{Nt}|}}(m_N) \cdot \frac{|x_t|}{\sqrt{2}}, \tag{49}$$

and the fraction of the right-hand side corresponds to the black curves in third row of plots of Figure 1 corresponding to the Majorana and Dirac cases. In this case we find that as long as $m_S > m_N$, the freeze-out is nicely described by the effective four-fermion operators. This is again in contrast to the $Z'$, where already close to $m_{Z'} = 2m_N$ the EFT limit is broken. Since $S$ is a $t$-channel mediator, the secluded DM annihilation into a pair of mediators is possible outside the EFT regime.

**LHC and other constraints on the UV-model**    We now explain which generic constraints in the EFT picture discussed in Section 2.2 and which model-specific constraints give rise to the excluded regions in Figure 6.

1. Perturbativity: again, we indicate in gray the region when the coupling becomes $|x_t| > \sqrt{4\pi} \approx 3.5$, where the theory becomes non-perturbative.

2. Collider production: collider signatures of leptoquarks are classified in Ref. [90]. The main production channel of our leptoquark is $gg \to SS^\dagger$ and does not depend on the coupling $x_t$, but is entirely determined by the strong gauge coupling $g_s$ and the leptoquark mass $m_S$. The value of the coupling $x_t$ is only relevant in so far as it is assumed that the decay width is sufficiently small to consider the leptoquarks to be produced on-shell. The most stringent bound on $m_S$ can be derived from a CMS stop search [91] (see Ref. [92] for a very similar ATLAS search). We use the analysis that assumes the stop decays into a neutral neutralino and a top, which essentially is our signature. In the case of $m_N \lesssim 400$ GeV this excludes $m_S \lesssim 1200$ GeV at 95% CL. We use the exact exclusion contours in Figure 6, they are shown in red.

3. Direct detection: in the Dirac case, the running-induced effective neutron coupling at nuclear scales leads to a bound from XENON1T, as discussed in Section 2.2. In the Majorana case, RG running induces no notable couplings and the two relevant interactions are the loop-induced effective Higgs and gluon couplings corresponding to $\mathcal{O}_{NH}^{(5)}$ and $\mathcal{O}_{gN}^{(7)}$. In the EFT region $m_S \gtrsim 3m_N$, the Higgs-coupling dominates [87]. Therefore, we again apply the more recent XENON1T bound on the effective Higgs operator shown in Figure 2 using the expression for the one-loop Higgs coupling Eq.(48). We find that the limits are not strong enough to probe the parameter space with $m_N \geq 160$ GeV shown in Figure 6. The bound on the WIMP-nucleon cross section would need to be improved by a factor of about 100 to cut into our parameter space, and a factor $10^4$ improvement on the WIMP-nucleon cross section would be required to rule out all of the mass range $m_N < 1000$ GeV. This means that we cannot expect XENONnT to produce notable constraints on the EFT region of this model [93].

4. Indirect detection: as can be concluded from the third row of Figure 1 (blue regions), current constraints from annihilation in dSphs are consistent with the full DM mass range that we consider, if only annihilation into fermion pairs is considered. From Ref. [65], however, we know that annihilations into gluon pairs are also relevant in the Majorana case for certain parameter configurations. As described in Section 2.2, the $NN \to gg$ annihilation channel becomes strong relative to the fermionic annihilation at the edge of our DM mass region near $m_N \sim 10^4$ GeV [87]. This additional contribution from gluons to the expected gamma ray signal is, however, by far not sufficient to probe the interesting region where $\Omega_N h^2 = 0.12$, as can be seen from the distance between the orange region and the black line in the bottom left plot of Figure 1. Therefore, indirect detection constraints calculated in the EFT at dimension six can be applied in this model.

To briefly summarize, there remains parameter space for a neutral fermion singlet that couples via scalar leptoquarks to $t$-quarks, consistent with an EFT description. While the leptoquark mass should exceed the TeV-scale, neutrino masses can be as low as $m_t$, in the Majorana case. In the Dirac case direct detection limits exclude DM masses up to 1002 GeV. Compared to a $Z'$ which couples only to $t_R$ and generates the same four-fermion EFT operator, we observe that the collider constraints are stronger in the leptoquark case.

## 3.3 Vector leptoquarks

As a third UV-completion we consider the vector leptoquark $U \equiv U_1'' \sim (3, 1, 2/3)$. It is also secluded from the SM in the sense that there exists one interaction with a SM fermion and a sterile neutrino. The simplified Lagrangian, focusing on interactions with the third generation only, reads

$$\mathcal{L}_{\text{LQ}} = m_U^2 U_\mu^\dagger U^\mu - \frac{1}{2}(U^{\mu\nu})^\dagger U_{\mu\nu} - i g_S \kappa \, U_\mu^\dagger T^a U_\nu G_{\mu\nu}^a \\ + x_b \, \overline{b_R} \gamma_\mu N_R \, U^\mu + x_b^* \, U^{\dagger\mu} \overline{N_R} \gamma_\mu b_R \,, \tag{50}$$

where

$$U_{\mu\nu} = D_\mu U_\nu - D_\nu U_\mu \qquad \text{and} \qquad D_\mu = \partial_\mu + i g_s T^a G_\mu^a \,. \tag{51}$$

The parameter $\kappa$ is fixed by the origin of the vector leptoquark and is either one or zero. Note that the leptoquark couples only to right-handed down-type quarks and right-handed sterile neutrinos. In addition, the coupling to gluons arises from the $SU(3)_c$-charge. As usual, we assume that $U$ is also odd under the $\mathbb{Z}_2$ symmetry, otherwise the second line in Eq.(50) is not allowed. This type of interaction is possible both for Majorana and Dirac sterile neutrinos and in both cases only the right-handed components will interact. As for the scalar leptoquark, small running-induced couplings to the down and strange quarks will be generated in a manner similar to the up-type quark case discussed in Section 3.2 [94]. We neglect them also in this case.

**Operator matching** Straightforwardly, for processes with momentum transfer well below $m_U$ the effective four-fermion interaction reads

$$\mathcal{L}_{\text{eff}} = \frac{|x_b|^2}{m_U^2} (\overline{b_R} \gamma^\mu N_R)(\overline{N_R} \gamma_\mu b_R) \,. \tag{52}$$

Again, this can be rephrased as the operator $\mathcal{O}_{Nb} \equiv C_{Nd}^{33}$, or

$$C_{Nb} = |x_b|^2 \,, \tag{53}$$

upon identifying $\Lambda = m_U$. Hence, the effective interaction mimics a vector-mediated $\overline{b}b \leftrightarrow \overline{N}N$ interaction and we can compare the scenario to a $Z'$ coupling only to $b_R$ and $N_R$ inducing the same operator.

**Relic density** We implement the full model via FeynRules [74] and calculate the DM relic density after freeze-out using MICROMEGAS [50–53]. In Figure 7 we plot the $m_N$-$m_U$ plane and fix at each point $|x_b|$ such that the observed DM relic density is obtained. We compare this to the EFT limit of the model described by the matching Eq.(53) together with $\Lambda = m_U$. This means that for each contour of constant $|x_b|$ we associate a dashed EFT contour with constant $\sqrt{|C_{Nb}|} = |x_b|$, where the $m_U$ coordinates are given by

$$m_U(m_N) = \frac{\Lambda}{\sqrt{|C_{Nb}|}}(m_N) \cdot |x_b| \,, \tag{54}$$

and the fraction of the right-hand side corresponds to the curves in the center row of Figure 1. We find that as long as $m_U > m_N$, the freeze-out is nicely described by the effective operators. This is again in contrast to the $Z'$, where already close to $m_{Z'} = 2m_N$ the EFT limit is broken. The reason is that for the $t$-channel mediator an on-shell annihilation has to go into a pair of mediators. The different positions of the lines for Dirac and Majorana neutrinos result from the different form of the annihilation cross sections, see Eqs.(11) and (12).

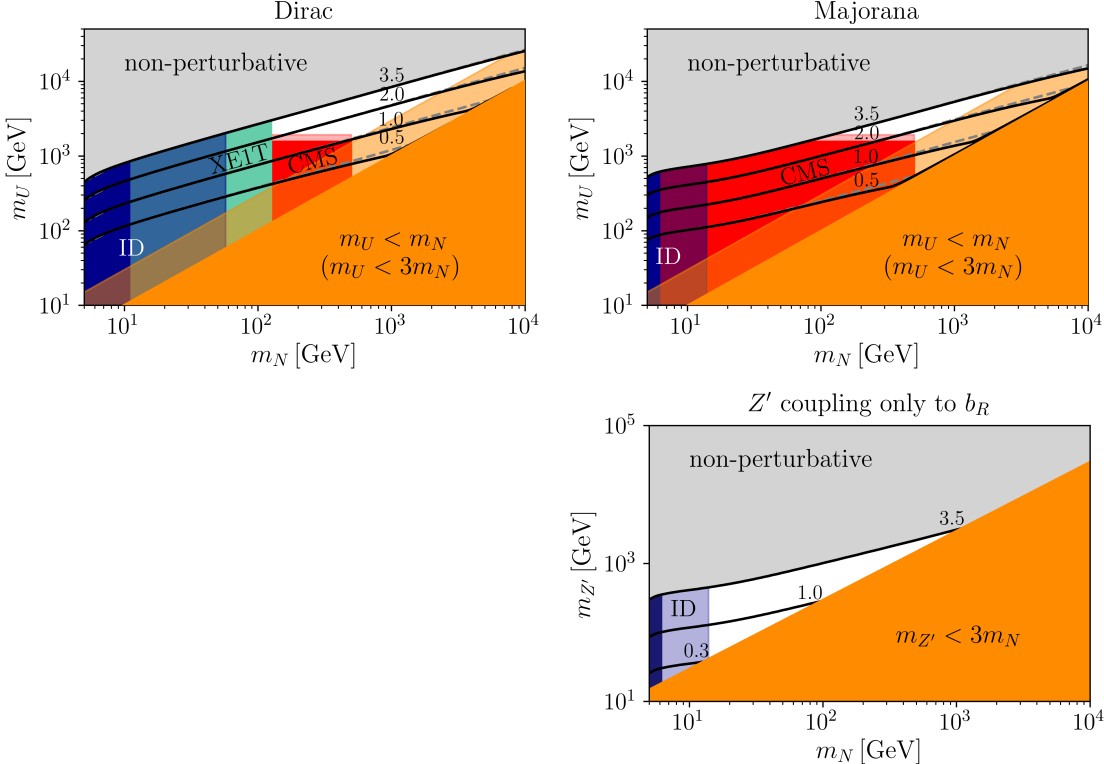

Figure 7: Upper: contours of $|x_b|$ in the $m_N$-$m_U$-plane producing the correct relic abundance. Dashed lines correspond to the EFT limit of the given coupling. Constraints from indirect and direct detection are very weak. CMS limits are shown in red, perturbativity of the coupling and validity of the EFT description ($m_S < 3m_N$ in orange, $m_S < m_N$ in light orange) are also indicated. Lower: same plot for $Z'$ models coupling only to the $B - L$ charge of right-handed top quarks. Contours of $g_X$ producing the correct relic abundance are shown in black, the $Z$-$Z'$ mixing parameter $\epsilon$ is zero.

**LHC and other constraints on the UV-model** As before, we combine generic constraints in the EFT picture, Section 2.2, with model-specific constraints in Figure 7.

1. Perturbativity: again, we indicate in gray the region when the coupling becomes $|x_b| > \sqrt{4\pi} \approx 3.5$, where the theory becomes non-perturbative.

2. Collider production: the main production channel of our vector leptoquarks is $gg \to UU^\dagger$, which does not depend on the coupling $x_b$, but is entirely determined by the strong gauge coupling and the leptoquark mass $m_U$ [88]. The value of the coupling $x_b$ is only relevant in so far as it is assumed that the decay width is sufficiently small to consider the leptoquarks to be produced on-shell. The most stringent bound on $m_U$ can be derived from the leptoquark search in CMS [95]. Assuming $m_N = 0$, the CMS analysis excludes $m_U \lesssim 1558$ GeV for $\kappa = 0$ and $m_U \lesssim 1927$ GeV for $\kappa = 1$ at 95% CL. Since a detailed exclusion contour of the $m_N$-$m_U$ parameter space is only given for a scalar leptoquark, we project the excluded mass range of $m_U$ given in the limit of massless invisible final states onto larger DM masses. Using the scalar leptoquark plot in Ref. [95] as guiding example, this appears to hold approximately up to mass of about 500 GeV. In Figure 7, the LHC limit obtained this way is shown in red ($\kappa = 0$) and light red ($\kappa = 1$). We note that a collider study of a vector leptoquark coupling to massive neutral fermions is

missing.

3. Indirect detection: as illustrated in the second row of Figure 1, current constraints from annihilation in dSphs can be used to set a lower bound on the DM mass in the case that the effective operator $\mathcal{O}_{Nb}$ induces the dominant contribution to DM annihilation. In Figure 7, we therefore include the lower bounds on $m_N$ including the band corresponding to the variation of the $J$-factors within their 68% CL intervals. Note again the different position of the indirect detection constraints for Dirac and Majorana fermions, which results from the different form of the annihilation cross sections, see Eqs.(11) and (12). In Section 2.2, we noted that the effective DM-gluon coupling may be relevant for indirect detection. Considering that for a scalar sbottom-like mediator we found no notable effect in the parameter region where the observed relic density is produced (see the orange band in the center-left plot of Figure 1), we expect that also in the vector case the effect can be neglected.

4. Direct detection: in the discussion in Section 2.2 we saw that the coupling to nucleons induced by RG running of the operator $\mathcal{O}_{Nb}$ is not yet tested with sufficient precision to constrain the model in the EFT region, see Figure 1. While there exists no discussion of the effective DM-Higgs and DM-gluon coupling mediated by vector leptoquarks, we boldly extrapolate from the scalar case that the explicit mass suppression weakens the direct detection limits to a level where they do not affect our EFT parameter space.

Again, we can summarize Figure 7 in that there exists parameter space for a neutral fermion singlet that couples via vector leptoquarks to $b$-quarks, in which a consistent EFT description is possible. For Dirac DM $m_N > 123$ GeV and TeV-ish leptoquarks are still allowed, while for the Majorana case, where direct detection constraints are weak, the sterile neutrino mass needs to exceed 77 GeV for $\kappa = 0$ and 123 GeV for $\kappa = 1$. The precision of these numbers would benefit from a dedicated evaluation of LHC constraints on vector leptoquark pair production with subsequent decays into $b$-quarks and massive dark fermions, as it has already done for scalar leptoquarks. Compared to a $Z'$ coupling only to $b_R$ and $N_R$, which leads to the same effective four-fermion operator, the collider constraints are stronger in the leptoquark case.

## 4 Conclusions

Given the energy scales involved, an EFT approach to weak-scale DM is still an attractive scenario. It can, for instance, reveal tensions between freeze-out production, direct detection, and indirect detection. The problem with a pure EFT approach is that dark matter mediators can appear on their mass shell either at colliders or at some time during the thermal history of the Universe. This is why we consider effective theories at the weak scale only as representative generalizations of UV-complete models. If we force the interpretation of LHC searches into a DMEFT framework, it immediately leads to tensions with the annihilation rate or observed relic density [27].

Sterile neutrinos as WIMP dark matter offer a natural way out of this when they couple only to third-generation fermions. Starting with a pure EFT approach we have demonstrated that the relic density can be generated while all constraints from direct and indirect detection, as well as lepton flavor universality are obeyed. This holds true when we couple the heavy neutrino to tau leptons, bottom quarks, and top quarks. LHC constraints are strongest when we produce the mediator on its mass shell, so we expect them to depend on the underlying model.

We have confronted three successful scenarios of our $\nu$DMEFT with a set of plausible UV-complete models. We studied a $(B-L)_3$ gauge extension of the Standard Model as well as

scalar and vector leptoquarks. In all cases, DM-mediator couplings to light fermions are absent at leading order. We found that all three models are well represented by the unifying EFT at dimension six, but that LHC searches naturally distinguish between the fundamental models. In general, the LHC reach for leptoquark UV-completions is larger, because of their unavoidable coupling to gluons and the corresponding pair production process. Search limits from supersymmetric squarks usually apply with minor modifications, even though this phase space approximation is less motivated for the vector leptoquark case.

In conclusion, we have demonstrated, with working examples, an EFT framework for the analysis of WIMPs which couple to the third generation SM fermions. The range of consistent scenarios within this framework is much larger than the possibilities we could consider here. For instance, including several operators allows individual annihilation cross sections to be smaller, which can, but need not necessarily, relieve constraints, as we have seen in Section 3.1. While currently there are numerous possibilities, we expect the EFT approach to be most useful when constraints from direct and indirect detection tighten or in the case of an observation in one of the channels. If the number of possible configurations of Wilson coefficients that fit observations is reduced, this could give clearer hints on how models need to be constructed. At present, the advantage compared to starting directly from explicit models lies in the fact that constraints from the relic density, direct detection, and indirect detection need to be calculated only once for a given hierarchy of Wilson coefficients. Namely, other models than the ones discussed in Section 3.2 and Section 3.3 which generate dominantly the DM-top or DM-bottom interaction could be constructed. It would then be sufficient to match the model parameters to the EFT to apply the previously calculated constraints. Of course, as for any finite EFT expansion, one could miss higher-dimensional operators with Wilson coefficients large enough to spoil the expansion in $1/\Lambda$. However, in the models we considered, these additional operators turned out to be negligible, which supports the expectation that such large higher-order terms are non-generic.

## Acknowledgments

We thank Tim Tait, Mathias Garny, Stefan Vogl and Thomas Hugle for helpful comments. I.B. is supported by the IMPRS-PTFS. T.P. is supported by the DFG under grant 396021762 – TRR 257 *Particle Physics Phenomenology after the Higgs Discovery*. W.R. is supported by the DFG with grant RO 2516/7-1 in the Heisenberg program.

## A  Details on the $Z' \to \tau\tau$ limit in the $(B-L)_3$ model

In this section, we summarize how we produced the exclusion curve of $Z' \to \tau\tau$ in Figure 5. We consider an EFT limit of the Lagrangian Eq.(22) along the following reasoning. At leading order, i.e. 0th order in $\epsilon$ and tree-level QCD, the only production channel of the $Z'$ resonance is $b\bar{b} \to Z'$. Therefore one needs to consider a 5-flavor parton distribution function to evaluate the $pp \to Z'$ production cross section. At the order $\epsilon^0$, we can ignore $Z$-$Z'$ mixing, and the interactions are simply determined by the $(B-L)_3$ charge of the respective fermions,

$$\mathcal{L}_{\text{NC}}^X = -\sum_f g_X q_X^f (\bar{f}\gamma^\mu f) Z'_\mu, \tag{A.1}$$

where $f = t, b, \tau, \nu_\tau, N$. Given these interactions, it is straightforward to calculate the decay width of $Z' \to \bar{f}f$, for $f$ a fermion with both chiralities, in the limit $m_{Z'} \gg m_f$,

$$\Gamma^X_{Z' \to \bar{f}f} = \frac{1}{12\pi} g_X^2 (q_X^f)^2 N_c^f m_{Z'} . \tag{A.2}$$

Therefore the total decay width, considering the channels $\bar{t}t$, $\bar{b}b$, $\tau^+\tau^-$, $\bar{\nu}_\tau \nu_\tau$, and $NN$ reads

$$\begin{aligned}
\frac{\Gamma^X_{Z'}}{m_{Z'}} &= \frac{1}{12\pi} g_X^2 \left[ (q_X^t)^2 \cdot 3 + (q_X^b)^2 \cdot 3 + (q_X^\tau)^2 + \frac{1}{2}(q^{\nu_\tau})^2 + \frac{1}{2}(q^N)^2 \right] \\
&= \frac{2}{9\pi} g_X^2 .
\end{aligned} \tag{A.3}$$

The interaction in Eq.(A.1) along with mass terms for $N$ and $Z'$ and the decay width in Eq.(A.3) was implemented as a Universal FEYNRULES Output (UFO) model file using FEYNRULES [74]. This model file was subsequently loaded into MADGRAPH [96], where we calculated the cross section for $pp \to \tau^+\tau^-$ to compare with the limits given by ATLAS [79].

# B  Details on $Z'$ with coupling only to $b_R$ or $t_R$

Here we discuss how we to test for constraints on a $Z'$ with effective couplings to only $N$ and $b_R$ or $t_R$, i.e.

$$\begin{aligned}
\mathcal{L}^1_{\text{int}} &= g_X \left( -N_R \gamma^\mu N_R + \frac{1}{3} b_R \gamma^\mu b_R \right) Z'_\mu , \\
\mathcal{L}^2_{\text{int}} &= g_X \left( -N_R \gamma^\mu N_R + \frac{1}{3} t_R \gamma^\mu t_R \right) Z'_\mu .
\end{aligned} \tag{B.1}$$

In Refs. [80, 81] limits on the production of a neutral scalar $H_2$ together with a $b$ pair or $t$ pair, respectively, have been presented in the form of cross section times branching into $H_2 \to \bar{b}b$ or $\bar{t}t$. We compare this product to the production cross section for $pp \to \bar{b}bZ'$ and $pp \to \bar{t}tZ'$ respectively, which we calculate using the interactions in Eq.(B.1) as before via MADGRAPH [96]. This has to be multiplied by the branching fraction into $b$ or $t$ pairs which, following Eqs.(A.2) with only right-handed quarks, is $1/4$. By comparing the cross sections for $m_{Z'} = m_{H_2}$, we find in both cases that the cross section limits are not yet strong enough to probe the model for $g_X \leq 3.5$. We note that the tiny red area that is the limit in the bottom plot of Figure 7 has been obtained for a branching ratio in $bb$ of 1, for a value of $1/4$ it would disappear in the plot.

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
