# Peer review of "Dark Matter EFT, the Third -- Neutrino WIMPs"

_SciPost Physics, doi:SciPost Phys. 10, 039 (2021)_

## Round 1 · Referee Report · Hai-Bo Yu (Referee 1) · 2020-11-2

Strengths

  1. This work explicitly shows that the WIMP is still a visible dark matter candidate. It helps the field clarify confusion about the validity of the WIMP scenario.
  2. The study is comprehensive.
  3. The presentation is clear and the paper is well written.

Report

In this paper, the authors proposed a dark matter scenario where sterile neutrinos couple to third generation fermions of the standard model. They took the approach based on effective field theory, and studied relic density, direct and indirect detection constraints. It shows that there is large viable parameter space for the sterile neutrino to be a thermal WIMP. The authors further proposed three UV models, and studied their LHC and lepton flavor violation constraints. The work is comprehensive and the presentation is clear.

I would like to recommend it for publication in SciPost after the authors address the following minor comments.

Requested changes

1. For direct detection constraints, most of the analysis is based on the limits from the XENON100 experiment. I guess this is because the one could directly use the results presented in the reference [39], arXiv:1705.02614, where the XENON100 collaboration interpreted their results using effective operators. Could the authors comment how the direct detection bounds shift after taking more recent XENON1T results? For simplicity, the authors may pick up one of the cases shown in Figure 1 .

2. In the first paragraph, page 8, clarify the form of the O1 operator.

3. In Figure 7, should the indirect detection bounds depend on the value of xb as well?

4. For the UV model based on the third generation (B-L) symmetry, it seems nontrivial to achieve the observed lepton flavor mixing angles. Please comment on it.

5. Both “Standard Model” and “SM” are used multiple times.

  • validity: high
  • significance: high
  • originality: good
  • clarity: high
  • formatting: excellent
  • grammar: excellent

Author:  Ingolf Bischer  on 2020-12-18  [id 1087]

(in reply to Report 1 by Hai-Bo Yu on 2020-11-02)
Category:
remark
answer to question

We thank the referee for their careful consideration and comments. Below we reply to requests/questions not obviously addressed by the list of changes in the revised manuscript. 3. The indirect detection bound in Figure 7 depends only on the DM mass, since the ratio of xb and mU is fixed in this plot to provide the correct relic abundance. This fixed ratio corresponds to the black line in Figure 1. The ratio xb /mU , however, determines not only the relic density, but also the indirect detection signal, since both are determined by the annihilation cross section proportional to |xb|^4 /mU^4. Therefore, for a fixed DM mass, fitting xb /mU to the relic density also fixes the indirect detection signal, which leads to the vertical exclusion line in Figure 7.

---

## Round 1 · Referee Report · Anonymous (Referee 2) · 2020-11-3

Strengths

  1. The article provides a systematic treatment of the class of models it aims to probe in a Dark Matter EFT.

  2. It proceeds to prove that those models are feasible, with multiple obvious UV completions which aren't otherwise ruled out.

  3. Its treatment of uncertainties arising from astrophysical unknowns in indirect detection is robust.

Weaknesses

  1. The article does not make clear how it treats points in parameter space which predict too little dark matter remaining from the given model, while treating the opposite case as an absolute bound. This isn't generic because the presence of a small additional annihilation channel can resolve an overabundance in a thermal DM scenario, but nothing can increase that abundance without a significantly different cosmological history.

  2. The article generally does not explore the possibility of multiple $\nu$DMEFT operators being simultaneously active, which limits the applicability to generic new physics which meets the flavor hypotheses defining the EFT, though it does still permit the set of UV completions studied by the authors.

  3. The flavor hypotheses are more weakly justified than the ideal; while they are explained as being necessary for the results of the article to be interesting in the context of LHC searches, it would be interesting to explore how deviations from them that map on to more motivated hypotheses in the flavor sector like Minimal Flavor Violation or a more robust U(2) flavor symmetric model which doesn't assume away entirely the light fermion couplings.

Report

This journal's acceptance criteria are, in my view, not met. This contribution is valid and valuable, but doesn't rise to the level of "groundbreaking," "breakthrough," or "new pathway." Physics Core may be more appropriate, after requested changes.

Requested changes

  1. Explain in detail the treatment adopted for development of direct and indirect detection bounds in the region studied where the DM abundance is predicted to be below the cosmologically known quantity.

  2. Explore multiple operators or additional annihilation pathways to open the parameter space currently 'closed' by overdensity concerns.

  3. Investigate a more complete flavor paradigm, at least providing solid approximations of the required smallness of couplings to light fermions in order for the EFT treatment to remain interesting, as defined by the authors (allowing for interesting EFT parameter space not strongly foreclosed by collider searches).

  • validity: high
  • significance: good
  • originality: good
  • clarity: high
  • formatting: perfect
  • grammar: perfect

Author:  Ingolf Bischer  on 2020-12-18  [id 1088]

(in reply to Report 2 on 2020-11-03)
Category:
remark

We thank the referee for their careful consideration and comments. Below we reply to requests/questions not obviously addressed by the list of changes in the revised manuscript. 2. We agree that the question of multiple operators or annihilation pathways would be of interest as a further application of the third-generation DMEFT. Therefore, some comments on the possiblity of multiple-operator scenarios to relieve overdensity bounds have been added, see Section 2.1, the discussions of direct and indirect detection in Section 3.1 and the conclusions. However, the scope of possiblities is very wide and in our understanding a topic for further publications. We have considered one example model with multiple operators and straightforward single-operator scenarios. These seem to us sufficient to demonstrate the extend to which the EFT can be applied. Further expansion of the set of considered scenarios, while interesting in general, to our mind would not significantly affect the conclusions of the manuscript at hand.

---

## Round 1 · Referee Report · Anonymous (Referee 3) · 2020-11-23

Strengths

1- The proposed EFT involving only third-generation fermions is interesting and the mapping from underlying UV completions involving sterile neutrinos provides valuable motivation. 2- The analysis is technically sound and includes all relevant constraints (although the XENON100 constraint is by now a bit outdated and anti-proton constraints from AMS-02 could provide valuable additional information).

Weaknesses

1- The treatment of the relic density constraint is unjustified. 2- The broader point on how EFTs should be combined with UV-complete models should address the case where several operators are induced simultaneously. 3- The presentation could be a bit less provocative and more balanced.

Report

The starting point of this paper is the observation that global fits of dark matter effective theories face the difficulty of how to include LHC constraints, which will either be overestimated (without EFT truncation) or conservative (with EFT truncation) and cannot capture signatures arising from the on-shell production of particles at the new-physics scale. It is pointed out that a possible resolution may be to focus on models where LHC constraints are weak, because dark matter couples dominantly to third-generation fermions. The authors then study such models both in the EFT approach and for specific UV completions and show how the two approaches are related. This is an interesting approach and the analysis is a valuable addition to the literature.

In addition, the authors try to make a broader point regarding the use of effective operators in global analyses of dark matter models. The idea is to use the EFT only for those constraints that can be reliably calculated within the effective approach (relic density, direct detection, indirect detection), while additional constraints (in particular from the LHC) should be calculated only for specific UV completions. However, when studying the specific UV completions, they identify various problems with this approach, related in particular to the fact that typically several operators are introduced at once. This makes it necessary to recompute the relic density for each UV completion and also include several operators introduced at higher dimension and/or loop level in the calculation of direct and indirect detection constraints. It therefore does not become clear what the benefit of the EFT approach is, i.e. in what way the analysis goes beyond the study of several UV-complete models of dark matter.

Nevertheless, once the technical issues mentioned below have been addressed and the presentation has been made clearer, the paper should be suitable for publication in SciPost.

Requested changes

1- The authors need to make clearer how they envision the DMEFT they introduce to be used in global fits, i.e. how they address the issues they mention in the introduction. Is the point simply that LHC constraints are conceivably sufficiently weak to perform a global analysis without including these constraints? Or is the point that a global analysis is never possible within an EFT and always needs to include constraints specific to UV-complete models. In the latter case, is the DMEFT only a calculational tool in order to make it easier to evaluate constraints on UV-complete models by mapping them onto effective operators? How does one then address the issue that different operators may interfere (in particular in the context of direct detection) and that the relic density constraint gets weaker rather than stronger when several operators are present?

2- I think the fact that the authors do not rescale their constraints based on the predicted relic abundance is a very significant shortcoming of the present analysis. In many DMEFT analyses this approach is justified by assuming a particle-antiparticle asymmetry, which would allow the reproduce the observed relic abundance even for very large annihilation cross sections. While such an asymmetry may be justified for Dirac neutrinos, it is impossible for Majorana neutrinos. In the absence of such an asymmetry, I think it will be very difficult to argue why points with Omega h^2 < 0.12 are treated as if they predicted Omega h^2 = 0.12, while points with Omega h^2 > 0.12 are treated as excluded.

3- While I understand how the models under consideration are inspired by models of sterile neutrinos, I find the name rather misleading, given that the defining property of sterile neutrinos, which is the active-sterile mixing induced by the \bar{l}N\tilde{H} Yukawa interaction, needs to be either absent or negligibly small in order for the dark matter particles to be stable. Do the dark matter particles in the present work have any other characteristic that distinguishes them from the fermionic dark matter particles commonly studied in the literature?

4- I currently find the presentation a bit unbalanced. I understand that the authors are deliberately provocative, but I would encourage them to be a bit more accurate in the following regards: a) The authors state twice that an EFT with no UV completion is utterly pointless''. While I agree with the sentiment, it is not clear to me how to use this requirement in practice, given that EFTs can also arise from strongly-interacting theories, for which the mapping onto the EFT may be non-trivial. It would be good if the authors could elaborate a bit on this point. b) The authors state that Ref. [23] constitutes alargely unsuccessful attempt'' to perform a global fit of the DMEFT. It seems to me that this is their own assessment, rather than the conclusion reached by the authors of the original study. This distinction should be made clearer and, ideally, elaborated on. c) The introduction is either oblivious or misleading regarding the work that has been done on DMEFTs. First of all, in spite of being hugely influential, Refs. [10,11] do not constitute the first attempts to construct an effective theory for dark matter and are predated by at least arXiv:0810.5557, arXiv:1002.4137 and arXiv:1003.1912. Second, the paper misses a large number of works carried out around 2011 at the height of the interest in DMEFTs. Then, the paper gives the impression that problems with the EFT approach in the context of collider physics were first pointed out in Ref. [13], even though these issues have been known since arXiv:1308.6799 and arXiv:1307.2253. Indeed, many attempts to address these issues by making sure that constraints from the LHC are always conservative were made subsequently, see e.g. arXiv:1402.1275 and arXiv:1405.3101. Likewise, it would be good to mention the very wide range of activities related to simplified models, which were introduced as the missing link between UV complete theories and EFTs. Finally, I want to particularly emphasize arXiv:1607.02475, which provides a comprehensive summary of this discussion and suggests ways to consistently apply the EFT framework. In addition to updating the bibliography, the authors should also consider changing the title of the paper, because I don't think they can justifiably argued to have made the third attempt to study DMEFTs.

  • validity: good
  • significance: good
  • originality: high
  • clarity: high
  • formatting: excellent
  • grammar: excellent

Author:  Ingolf Bischer  on 2020-12-18  [id 1086]

(in reply to Report 3 on 2020-11-23)
Category:
remark

We thank the referee for their careful consideration. Below we comment on some of the requests unless obvious from the list of changes in the revised version.

2- We agree that the regions of parameter space where Ωh 2 < 0.12 do not provide a consistent model of dark matter. In the first version of the manuscript this region is shown as unexcluded in the EFT plots Fig. 1 and 3 with the intention to show in-principle available parameter space, for which to be able to explain observations, however, there needs to be another DM particle filling the density gap. For clarity of presentation, we have updated the plots to treat the parameter space where Ωh^2 < 0.12 and Ωh^2 > 0.12 equally. Both are indeed excluded for the simple one-operator EFTs with one sterile neutrino as dark matter.

On the point of rescaling constraints based on the predicted relic abundance we would like to emphasize that the appropriate rescaling of the couplings to Ωh^2 = 0.12 has been performed in all models of Section 3 and their corresponding figures. Concering the EFT plots in Figure 1, we think that showing the “unrescaled” constraints in this form is of some value to the reader for the following reasons. First, it allows us to visualize constraints which not yet probe the EFT parameter space such as the indirect detection constraints in the tt case and as such give a visual measure of how far away from probing the parameter space they currently are. Second, it illustrates the shape of the bounds, which could be of interest if there is an independent mechanism for generating the relic density not captured by the dimension-6 operators. Meanwhile the relevant constraints for the single-operator scenarios can still be readily read off from the intersection of exclusion ranges and the relic abundance line.

4- a) We have addressed this request in the introduction. b) We changed the embedding of this reference to more accurately reflect its conclusions about a lower bound on the DM mass. c) We have extended the bibliography with the suggested literature and some further references. The claim on having made the third attempt to DMEFTs has been removed, while we consider the name referencing the third generation of fermions fitting.

---

## Round 2 · Referee Report · Hai-Bo Yu (Referee 1) · 2020-12-22

Report
In the revised draft, the authors have addressed all my comments. The work meets one of the expectations, i.e., "Present a breakthrough on a previously-identified and long-standing research stumbling block."
It also meets all the general acceptance criteria.
It also meets all the general acceptance criteria.

---

## Round 2 · Referee Report · Anonymous (Referee 3) · 2021-1-30

Report
The rather unspecific list of changes makes it difficult to assess in detail whether all my comments have been addressed. However, I have no reason to believe that they haven't and therefore recommend the paper for publication.

---

## Round 2 · List of Changes

- Direct detection constraints on Dirac DM have been recalculated employing the more recent results from XENON1T.
- The assumptions made when calculating direct detection and indirect detection constraints have been clarified, namely saturated relic density, and how to interpret the plots. Figures 1 and 3 now show over- and underdensity equally.
- More emphasize is put on the fact that the relic density, direct detection and indirect detection constraints are consistently described by the dimension-6 level EFT in our considered parameter space and that effects of higher-order operators turn out negligible.
- The form of the non-relativistic operator O1 and its role in generating spin-independent scattering has been clarified.
- The contextualisation within the existing body of literature on DMEFTs has been expanded in the introduction.
- Comments in the conclusions on how we envision the use of the third-generation DMEFT have been added.
- The equivalence between the nuDMEFT with the assumptions made in the manuscript and usual DMEFT has been pointed out more prominently.
- Comments on the possibility to relax constraints by considering multiple operators have been added.
- Comments and references addressing the consistency of the models with the SM flavor structure have been added.
- The NR operators for direct detection generated in the case of Majorana DM have been corrected to numbers 4, 8 and 9 from 5, 9, and 10.
- Minor formatting adjustments for some equations have been made.

---

## Editorial Decision

published